# Majority Bit-aware Watermarking for Large Language Models

## Abstract

The growing deployment of Large Language Models (LLMs) in real-world applications has raised concerns about their potential misuse in generating harmful or deceptive content. To address this issue, watermarking techniques have emerged as a promising solution by embedding identifiable binary messages into generated text for origin verification and misuse tracing. While recent efforts have explored multi-bit watermarking schemes capable of embedding rich information such as user identifiers, they typically suffer from the fundamental trade-off between text quality and decoding accuracy: to ensure reliable message decoding, they have to restrict the size of preferred token sets during encoding, yet such restrictions reduce the quality of the generated content. In this work, we propose MajorMark, a novel watermarking method that improves this trade-off through majority bit-aware encoding. MajorMark selects preferred token sets based on the majority bit of the message, enabling a larger and more flexible sampling of tokens. In contrast to prior methods that rely on token frequency analysis for decoding, MajorMark employs a clustering-based decoding strategy, which maintains high decoding accuracy even when the preferred token set is large, thus preserving both content quality and decoding accuracy. We further introduce MajorMark$^+$, which partitions the message into multiple blocks to independently encode and deterministically decode each block, thereby further enhancing the quality of watermarked text and improving decoding accuracy. Extensive experiments on state-of-the-art LLMs demonstrate that our methods significantly enhance both decoding accuracy and text generation quality, outperforming prior multi-bit watermarking baselines. The code of the proposed methods is available here for review.

## 1 Introduction

The rapid advancement and widespread deployment of Large Language Models (LLMs) have fundamentally reshaped numerous domains, from education and customer support to sophisticated content generation, owing to their capacity for highly fluent and contextually relevant text. However, this powerful generative capability simultaneously introduces considerable security and ethical challenges. Malicious users can leverage LLMs to produce harmful content, such as fake news, phishing emails, and fraudulent reviews (Qu et al., 2024; Zhang et al., 2025). To mitigate these risks, developing watermarking techniques is essential. By embedding an imperceptible yet verifiable signal within generated text, watermarking enables effective origin tracing and attribution of misuse (Liu et al., 2024c; Wu et al., 2025; Pan et al., 2024).

A typical text watermarking method comprises an *encoder* that embeds a specific message (such as a yes/no signal) into the generated content and a corresponding *decoder* designed to extract this message. This work focuses on a common watermarking scenario (Yoo et al., 2024; Qu et al., 2024; Jiang et al., 2025), where a cloud-based LLM-as-a-Service provider adopts a watermarking mechanism to uniquely mark each LLM-generated output for a user prompt. The decoder can then be applied to unverified text to recover the embedded message. The objective of watermarking is to ensure high decoding accuracy while preserving the utility of the generated content.

The first LLM watermarking scheme, KGW (Kirchenbauer et al., 2023), was designed to encode a zero-bit message–a yes/no signal indicating that an LLM generated the text. During encoding, at each token generation step, KGW pseudo-randomly selects a *green list* $\mathcal{G} \subseteq \mathcal{V}$ from the vocabulary

and strategically boosts the logits of the tokens in $\mathcal{G}$ by a watermarking bias $\delta$. This increases their probability of being sampled as the next token. A larger $\delta$ produces a stronger watermark but often results in noticeable degradation in the quality of the generated text. During decoding, KGW reconstructs the green list for each token within the target text and computes the proportion of tokens that fall into their respective green lists. If this proportion is statistically significant (measured via a $z$-test), the target text is verified as watermarked and, consequently, generated by an LLM.

However, zero-bit watermarking methods are inherently limited when the service providers need to embed more complex information, such as user identifiers, model versions, or other metadata. To address this, recent studies (Yoo et al., 2024; Li et al., 2024; Wang et al., 2024; Fernandez et al., 2023; Qu et al., 2024; Jiang et al., 2024; Yoo et al., 2023; Jiang et al., 2025) have explored multi-bit watermarking, designed to embed a binary message $\mathbf{m}$ of length $b \geq 2$. In some such schemes (Li et al., 2024; Wang et al., 2024; Fernandez et al., 2023), a unique watermark pattern is designed for each of the $2^b$ possible messages. This necessitates a brute-force search across all $2^b$ message candidates during decoding to determine which, if any, was embedded, a process that quickly becomes computationally infeasible as message length $b$ increases. To improve decoding efficiency, advanced methods such as MPAC (Yoo et al., 2024) and RSBH (Qu et al., 2024) divide the message into smaller blocks, each of which is independently encoded into the text. During decoding, only the candidate message for each block needs to be identified, thereby reducing the overall search space. However, existing multi-bit watermarking literature has primarily focused on decoding accuracy and efficiency, while the utility of watermarked text in encoding is less discussed. There are two key factors that determine the utility of watermarked text: the green list ratio, defined as $\gamma = |\mathcal{G}|/|\mathcal{V}|$, and the watermarking bias for boosting green lists' logits. These are typically treated as hyperparameters in existing methods.

In this work, we focus on studying the impact of the green list ratio $\gamma$ with a fixed watermarking bias $\delta$. We find that, for existing methods, a larger $\gamma$ leads to improved utility of the watermarked text, as a broader green list offers more choices for contextually suitable tokens during generation. In the extreme case where $\gamma = 1.0$ (i.e., the entire vocabulary constitutes the green list), the logit distribution remains unchanged after boosting. Conversely, a larger $\gamma$ undesirably leads to reduced decoding accuracy. This is because the decoding process in existing methods relies on counting the frequency of tokens that appear in their green lists. As a result, a larger green list provides a weaker statistical signal above random chance, making it harder to distinguish watermarked text reliably. Based on this observation, we propose a novel watermarking paradigm that (1) *eliminates the need for tuning the green list ratio $\gamma$*, (2) *improves the utility of watermarked text*, and (3) *enhances decoding accuracy*. Specifically, our method leverages the inherent dominant occurrence of the majority bit in the message to guide the construction of the green list $\mathcal{G}$, ensuring a guaranteed green list ratio of $\gamma \geq 0.5$. Unlike existing approaches that rely on counting how frequently generated tokens fall into $\mathcal{G}$ for decoding, we avoid such frequency-based heuristics. Instead, we recover the embedded message by analyzing the occurrence of tokens across predefined vocabulary shards, enabling more accurate decoding. Our key contributions are summarized as follows:

- We propose a majority bit-aware watermarking method, MajorMark, which leverages the majority bit in the $b$-bit binary message to generate the green list. This design guarantees that $\gamma \geq 0.5$, and in expectation $\mathbb{E}_{\mathbf{m}}[\gamma] \approx 0.5 + 1/\sqrt{2\pi b}$, leading to improved utility of watermarked text and eliminating the hyperparameter $\gamma$.

- Unlike existing methods, MajorMark eliminates the decoding's reliance on counting the tokens in green lists, so that the size of the green list does not have a direct impact on the decoding accuracy. Instead, it performs clustering over token occurrences within predefined shards, enabling more robust and fine-grained message recovery.

- We further introduce MajorMark$^+$, an enhanced variant of MajorMark. MajorMark$^+$ divides the message into $r$ blocks and encodes/decodes each block independently. It adopts MajorMark's encoding scheme for each block, resulting in a better text utility with $\mathbb{E}_{\mathbf{m}}[\gamma] \approx 0.5 + 1/\sqrt{2\pi(b/r)}$. During decoding, MajorMark$^+$ deterministically reconstructs each block by inferring both the majority bit and its occurrences, eliminating the need for clustering and thus enhancing decoding reliability.

- We conduct an extensive evaluation of our methods using state-of-the-art open-source LLMs. The results demonstrate that our methods consistently achieve higher decoding accuracy and better quality of generated text than existing multi-bit watermarking approaches.

## 2 EXISTING WORKS AND KEY OBSERVATIONS

**Large Language Models.** An LLM $f$ is an autoregressive model that performs the next-token prediction task. Specifically, given an input prompt $\mathbf{x}_p$, the model generates a sequence of tokens over a predefined vocabulary $\mathcal{V}$, which contains all permissible tokens for constructing natural language responses. Specifically, at the $t$-th ($t = 1, 2, \ldots, T$) generation step, the model takes as input the prompt $\mathbf{x}_p$ and the previously generated tokens $\mathbf{x}_{:t-1} = \{x_1, x_2, \ldots, x_{t-1}\}$, and outputs a logits vector $\ell^t = (\ldots, f(v \mid \mathbf{x}_p, \mathbf{x}_{:t-1}), \ldots)$, where $f(v \mid \mathbf{x}_p, \mathbf{x}_{:t-1})$ represents the logit for a specific token $v \in \mathcal{V}$. These logits are then passed through a $\texttt{softmax}(\cdot)$ function to produce a probability distribution over $\mathcal{V}$, from which the current token $x_t$ is sampled. This process is repeated autoregressively until a termination condition (e.g., a maximum length constraint) is met, yielding an output sequence $\mathbf{x}_g$ of length $T$.

**Zero-bit Watermarking in LLMs.** The first zero-bit watermarking method, KGW, was introduced by Kirchenbauer et al. (2023). Specifically, during the generation of the $t$-th token, after obtaining the logit vector from the LLM $f$, a pseudo-random seed is computed using a hash function that takes as input a secret key $k$ and the preceding token $x_{t-1}$. This seed is then used to deterministically permute and partition the vocabulary $\mathcal{V}$ into two disjoint subsets: the green list $\mathcal{G}$ and the red list $\mathcal{R}$. A positive watermarking bias $\delta$ is added to the logits of tokens in $\mathcal{G}$, thereby increasing their sampling probability after $\texttt{softmax}(\cdot)$. A larger $\delta$ introduces a stronger watermark but significantly alters the token sampling probabilities, which can substantially degrade the quality of the watermarked text. This process is applied at every token generation step, yielding a watermarked sequence $\mathbf{x}'_g$. To detect the watermark in a given unverified text $\mathbf{x}$, a decoder reconstructs the green list for each token in $\mathbf{x}$ and performs a $z$-test on the frequency of tokens that fall within their respective reconstructed green lists. A statistically significant excess over the expected frequency indicates the presence of the watermark. Several subsequent works have improved this zero-bit watermarking method to achieve higher decoding accuracy and better utility of the watermarked text (Kirchenbauer et al., 2024; Kuditipudi et al., 2024; Wang et al., 2025; Chang et al., 2024; Giboulot & Furon, 2024; Liu et al., 2024b; Piet et al., 2025; Christ et al., 2024; Munyer et al., 2024). Several works also introduce zero-bit watermarking methods that are robust under post-generation text editing (Liu et al., 2024a;b; Hou et al., 2024; Dabiriaghdam & Wang, 2025).

**Multi-bit Watermarking in LLMs.**
Multi-bit watermarking aims to encode and decode a $b$-bit binary message $\mathbf{m} \in \{0,1\}^b$ into and from the generated text. Multi-bit messages can carry richer information, such as the user's identity, the model version, or a timestamp. To embed $\mathbf{m}$ into LLM-generated text, CTWL (Wang et al., 2024) and CycleShift (Fernandez et al., 2023) leverage $\mathbf{m}$ to calculate the pseudo-random seed during encoding. However, their decoding processes require brute-force enumeration over all $2^b$ possible message values to find the best candidate, rendering them computationally infeasible for large $b$. A more efficient

Table 1: Comparison of representative multi-bit LLM watermarking methods. $\mathbb{E}_{\mathbf{m}}[\gamma]$ denotes the expected green list ratio over all possible messages. $\times |\mathbf{x}|$ indicates that the decoding process requires how many enumerations over all tokens in the input text $\mathbf{x}$. "Eff.?" reflects whether the decoding procedure is computationally efficient.

| Method | Encoding $\mathbb{E}_{\mathbf{m}}[\gamma]$ | Decoding $\times\|\mathbf{x}\|$ | Eff.? |
|---|---|---|---|
| CTWL (Wang et al., 2024)[†] | − | $2^b$ | ✗ |
| CycleShift (Fernandez et al., 2023) | $= 0.25$ | $2^b$ | ✗ |
| DepthW (Li et al., 2024) | $= 0.50$ | $2^b$ | ✗ |
| RSBH (Qu et al., 2024)[‡] | $= 0.50$ | $2^d$ | ✓ |
| MPAC (Yoo et al., 2024) | $= 0.25$ | $1$ | ✓ |
| MajorMark (Ours) | $\approx 0.50 + 1/\sqrt{2\pi b}$ | $2$ | ✓ |
| MajorMark[+] (Ours)[♯] | $\approx 0.50 + 1/\sqrt{2\pi (b/r)}$ | $b - r$ | ✓ |

[†] CTWL determines $\gamma$ dynamically by a proxy LLM.
[‡] In RSBH, $d$ denotes the number of bits in a message block.
[♯] In MajorMark[+], $r$ with $r < b$ denotes the number of message blocks.

method, MPAC (Yoo et al., 2024), divides the message $\mathbf{m}$ into $r$ blocks $\{\mathbf{m}_1, \mathbf{m}_2, \ldots, \mathbf{m}_r\}$, where each block is $d$ bits long. During generation, one message block is encoded per token generation step. Specifically, for each block $\mathbf{m}_i$, the encoder partitions the vocabulary into $2^d$ disjoint shards and selects the $[\mathbf{m}_i]_{10}$-th shard as the green list for that step, where $[\cdot]_{10}$ denotes the decimal (base-10) representation of the binary block. For decoding, MPAC identifies the shard with the highest token count at each step, interprets its index as the base-10 value of the corresponding message block, and converts it back to a $d$-bit binary string. The full message is then reconstructed by concatenating all recovered blocks. However, MPAC fixes $\gamma = 1/2^d$ (e.g., $\gamma = 0.25$ for $d = 2$ in their default setting), which significantly degrades the utility of the generated text. To address this, RSBH (Qu et al., 2024) improves MPAC by computing hash seeds using $[\mathbf{m}_i]_{10}$, choosing a larger

green list size of $\gamma = 0.5$. This enhancement improves utility but increases decoding complexity by a factor of $2^d$. A recent method, StealthInk (Jiang et al., 2025), adopts MPAC's block-wise encoding strategy by directly adjusting the sampling probabilities of specific tokens, thereby preserving the quality of the watermarked text. However, StealthInk suffers from low decoding accuracy. We summarize the decoding complexity of representative methods in Table 1.

**Trade-off between Text Utility and Decoding Accuracy in LLM Watermarking.** Existing multi-bit watermarking methods typically necessitate careful selection of the green list ratio $\gamma$ to balance the trade-off between the utility of watermarked text and the accuracy of decoding. Specifically, due to the shift invariance of the $\texttt{softmax}(\cdot)$ function, adding the bias to a larger portion of logits (i.e., using a larger $\gamma$) better preserves the original token probability distribution, thereby maintaining the quality of the generated sequence. However, since these methods rely on counting the frequency of watermarked tokens that fall within the recovered green list $\mathcal{G}$, a larger $\gamma$ weakens the watermark behavior, reducing the distinctiveness of the perturbation and making decoding less accurate. Conversely, using a smaller $\gamma$ introduces stronger distortions to the output distribution, thereby improving the ability to decode the embedded message. Yet this comes at the cost of generation quality: high-probability (i.e., desirable) tokens may fall outside $\mathcal{G}$ and be suppressed due to the bias, while low-probability tokens within $\mathcal{G}$ may be unintentionally amplified and incorrectly sampled. This imbalance can significantly degrade the quality of $\mathbf{x}'_g$.

To empirically illustrate the role of $\gamma$ in shaping this trade-off, we conduct experiments using RSBH (Qu et al., 2024) with LLaMA-2-7B (Touvron et al., 2023), varying the green list ratio $\gamma$ from 0.3 to 1.0 under message length $b = 64$. Note that $\gamma = 1.0$ corresponds to the extreme case where watermarking is ineffective. We report both the bit accuracy (i.e., the proportion of bits correctly decoded from the watermarked text) and the perplexity of the watermarked text in Figure 1, under different bias settings. As expected, increasing $\gamma$ consistently improves text quality, as reflected by

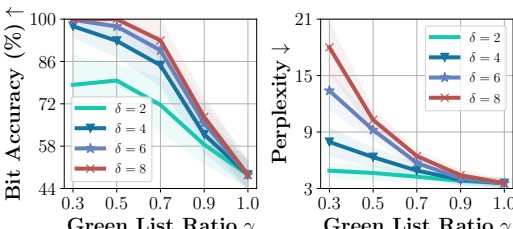

Figure 1: Impact of green list ratio $\gamma$ on the utility of watermarked text (right) and the decoding accuracy (left).

lower perplexity across all bias levels, but at the cost of reduced bit accuracy. These results empirically demonstrate the trade-off between text utility and decoding accuracy influenced by $\gamma$. We summarize the green list ratios $\gamma$ used in prior methods in Table 1. All of them set $\gamma \leq 0.5$, by following previous works (Wang et al., 2024; Fernandez et al., 2023; Li et al., 2024; Qu et al., 2024) or relying on empirical tuning (Yoo et al., 2024).

In the following sections, we introduce a novel majority bit-aware encoding method that embeds the message into the identity of token shards rather than their cumulative frequency. Crucially, this approach decouples the strength of the watermarking signal from the green list size, thereby allowing a large green list to ensure high text quality. Consequently, the decoding process shifts to recovering the identity of the green shards used during encoding, which achieves high decoding accuracy even when a large green list is used in encoding.

## 3 MAJORMARK AND MAJORMARK$^+$

### 3.1 PROBLEM FORMULATION

Given a language model $f$, a user prompt $\mathbf{x}_p$, and a $b$-bit binary message $\mathbf{m} \in \{0, 1\}^b$, the goal of multi-bit watermarking is to embed $\mathbf{m}$ into the generated text to produce a watermarked output $\mathbf{x}'_g$. This process is performed by an encoder function $\texttt{Enc}(\cdot)$: $\mathbf{x}'_g = \texttt{Enc}(f, \mathbf{x}_p, \mathbf{m})$, where $\mathbf{x}'_g$ should remain fluent and semantically consistent with the unwatermarked output $\mathbf{x}_g$. The embedded message can be extracted using a decoder $\texttt{Dec}(\cdot)$ to obtain $\mathbf{m}' = \texttt{Dec}(\mathbf{x}'_g)$. Formally, the objective of multi-bit watermarking can be formulated as the following constrained optimization problem:

$$\max \quad \Pr\Big[\mathbf{m}' = \mathbf{m} \mid \mathbf{m}' = \texttt{Dec}(\texttt{Enc}(f, \mathbf{x}_p, \mathbf{m}))\Big]$$
$$\text{s.t.} \quad \texttt{Quality}(\mathbf{x}'_g \mid \mathbf{x}_p) \geq \tau,$$

(a) MajorMark encoding process at generation step $t$.

(b) MajorMark$^+$ block-wise encoding process ($r = 2$) at generation step $t$.

Figure 2: An overview of the majority bit-aware encoding process of MajorMark and MajorMark$^+$.

where $\texttt{Quality}(\cdot)$ quantifies the quality of generated text, and $\tau$ is a minimum quality threshold (often chosen to be close to the quality of $\mathbf{x}_g$). In practice, conditional perplexity $\texttt{PPL}(\cdot)$ is widely used as a quality metric. This formulation captures the key goal of watermarking: to maximize the probability of exact message recovery while ensuring negligible degradation in text quality.

## 3.2 MAJORMARK

**Majority Bit.** We first define the majority bit $\lambda$ of a multi-bit message in Definition 1.

**Definition 1** (Majority Bit). *Given a binary message* $\mathbf{m} \in \{0,1\}^{b \geq 2}$, *let* $h_0$ *and* $h_1$ *denote the number of occurrences of bit 0 and 1 in* $\mathbf{m}$, *respectively. The majority bit* $\lambda \in \{0,1\}$ *of* $\mathbf{m}$ *is defined as* $\lambda = 1$ *if* $h_1 \geq h_0$, *and* $\lambda = 0$ *otherwise.*

For example, given a message $\mathbf{m} \in \{0,1\}^6$ such as $\mathbf{m} = 110111$, the majority bit is $\lambda = 1$ because $h_1 = 5$ and $h_0 = 1$. In cases where the occurrences are equal (i.e., $h_0 = h_1 = b/2$), we define the majority bit as $\lambda = 1$ by default. Crucially, the majority bit for any messages appears at least $\lceil b/2 \rceil$ times. This property provides a stable heuristic for message encoding. Our proposed method, MajorMark, leverages the dominance of the majority bit to guide the generation of the green list.

**Majority Bit-aware Encoding.** We present an overview of MajorMark's majority bit-aware encoding process at the $t$-th token generation step of LLM in Figure 2a. Specifically, prior to the generation process of the LLM $f$, MajorMark first identifies the majority bit $\lambda$ of the $b$-bit binary message $\mathbf{m}$ to be embedded. During the generation of the $t$-th token, after obtaining the logits $\ell^t$ from $f$, MajorMark performs the following steps: ① Computes a pseudo-random seed $s$ based on a hash function[1]; ② Applies the seed $s$ to permute the vocabulary $\mathcal{V}$, yielding a permuted vocabulary $\mathcal{V}'$; ③ Evenly partitions $\mathcal{V}'$ into $b$ disjoint shards $[\mathcal{V}_1, \mathcal{V}_2, \ldots, \mathcal{V}_b]$ and forms the green list by unioning the shards of the majority bit, such that $\mathcal{G} = \cup_{i:m_i=\lambda} \mathcal{V}_i$, where $m_i$ denotes the $i$-th bit of $\mathbf{m}$; ④ Uses the same seed $s$ to permute $\ell^t$, yielding permuted logits $\ell^{t,\prime}$ and adds a bias $\delta$ to the logits of tokens in $\mathcal{G}$; ⑤ Applies the $\texttt{softmax}(\cdot)$ function to the biased logits, yielding a new probability distribution from which the current token $x_t$ is sampled. Our majority bit-aware encoding strategy ensures a minimum green list ratio of $\gamma \geq 0.5$, thereby preserving the utility of watermarked text. In the following, we derive the theoretical guarantee for $\gamma$.

**Theorem 1** ($\gamma$ of MajorMark). *For any message* $\mathbf{m} \in \{0,1\}^{b \geq 2}$, *the green list* $\mathcal{G}$ *constructed via MajorMark satisfies* $|\mathcal{G}| \geq 0.5|\mathcal{V}|$, *i.e.,* $\gamma \geq 0.5$. *Moreover, for random $b$-bit binary messages, we have* $\mathbb{E}_{\mathbf{m}}[\gamma] \approx 0.5 + 1/\sqrt{2\pi b}$.

*Proof.* The detailed proof is in Appendix A.15. □

**Remark 1.** *MajorMark achieves a strictly larger expected* $\gamma$ *(*$\mathbb{E}_{\mathbf{m}}[\gamma] > 0.5$*) than existing methods for any finite $b$, indicating improved text utility. As* $b \to \infty$*, the expectation* $\mathbb{E}_{\mathbf{m}}[\gamma]$ *gradually converges to 0.5, making the improvements less significant. Note that we derive the expected* $\gamma$ *by assuming each bit is an independent random variable following a* Bernoulli$(0.5)$ *distribution. However, there are two extreme cases* $\mathbf{m} = \mathbf{0}^b$ *and* $\mathbf{m} = \mathbf{1}^b$ *that will result in* $\gamma = 1.0$*, thereby rendering*

[1]The design of the hash function is detailed later.

*the watermarking ineffective. In practice, these two special cases can be explicitly disallowed during encoding, and this has a negligible impact on the expected value of $\gamma$ when $b$ is large.*

**A Balanced Hash Function.** We revisit the design of MajorMark's hash function, which is crucial for both encoding and decoding. Prior works (Kirchenbauer et al., 2023; Yoo et al., 2024) typically compute the seed $s$ as $\texttt{Hash}(k, x_{t-1})$, where $k$ is a secret key and $x_{t-1}$ is the previous token. However, frequent tokens (e.g., *the*, *is*, *to*) appear disproportionately in generated text, causing certain seeds, and thus token-to-shard mappings, to repeat (Qu et al., 2024). This leads to repetitive green lists $\mathcal{G}$, over-sampling of specific tokens, and degraded text quality. To address this, we follow (Li et al., 2024) and extend the hash input to include the second-to-last token $x_{t-2}$, using $\texttt{Hash}(k, x_{t-1}, x_{t-2})$. This yields more diverse token-to-shard mappings. We further incorporate the majority bit $\lambda$, forming $\texttt{Hash}(k, x_{t-1}, x_{t-2}, \lambda)$, which increases seed diversity and facilitates recovery of $\lambda$ during decoding, as detailed below.

**Clustering-based Decoding.** The message decoding process takes as input a generated text $\mathbf{x}$ of length $T$ and produces the decoded message $\mathbf{m}'$. Specifically, for each token $x_t \in \mathbf{x}$ with $t = 3, 4, \ldots, T$, MajorMark reconstructs the token-to-shard mapping for $x_t$ used in encoding. We discard the first two tokens $x_1$ and $x_2$ from decoding, as computing their token-to-shard mappings requires access to the prompt tokens in $\mathbf{x}_p$, which are unavailable during decoding. To reconstruct the token-to-shard mapping for $x_t$, we compute the seed $s$ using hash $\texttt{Hash}(k, x_{t-1}, x_{t-2}, \lambda)$. We then determine the shard to which $x_t$ belongs and increment its corresponding count. After processing all tokens, we obtain a shard-wise token occurrence count that reflects the frequency of tokens belonging to each shard. Based on this occurrence count, we apply a clustering algorithm (e.g., $\texttt{KMeans}$ (McQueen, 1967) with $K = 2$) to partition the shards into two clusters. The cluster with the higher average token count, denoted by $\mathcal{C}$, is interpreted as corresponding to the majority bit $\lambda$, as the logits of tokens in these shards were positively perturbed by $\delta$ during encoding. The message bits are then recovered by setting $m_i = \lambda$ for each shard index $i \in \mathcal{C}$ and $m_i = 1 - \lambda$ for those not in the cluster (i.e., $i \notin \mathcal{C}$). We provide the theoretical justification for using clustering-based decoding and analyze decoding errors in Appendix A.17.

One remaining challenge in MajorMark's decoding process is determining the correct value of $\lambda$. Recall that we explicitly include $\lambda$ as the input of the hash function $\texttt{Hash}(k, x_{t-1}, x_{t-2}, \lambda)$. This design allows the recovery of $\lambda$. The key intuition is that only the correct seed $s$ can reproduce the token-to-shard mappings used during encoding. Therefore, in the decoding phase, we compute the shard-wise token occurrence counts for each possible value of $\lambda$ (i.e., 0 or 1). The correct value of $\lambda$ is determined as the one that produces a more skewed occurrence distribution (e.g., exhibiting a larger standard deviation), whereas the incorrect value yields a nearly uniform distribution, as illustrated in Figure 3. This

**Message $\mathbf{m} = 110111 \Rightarrow$ Majority bit $\lambda = 1$**

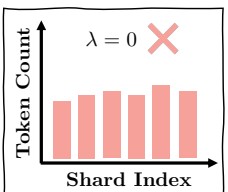 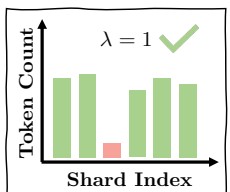

Figure 3: Occurrence distributions for candidate $\lambda$ values (0 or 1) during decoding.

approach enables reliable recovery of the embedded message with only a one-time additional enumeration over $\mathbf{x}$. Unlike prior methods that rely on counting how often tokens fall into the green list, MajorMark's decoding function eliminates this dependency. This enables accurate message decoding under a large green list setting. The detailed encoding and decoding algorithms of MajorMark and the implementation of the hash function are presented in Appendix A.13.

### 3.3 MAJORMARK$^+$: AN IMPROVED METHOD

We now introduce MajorMark$^+$, an enhanced version of MajorMark that further improves both the quality of watermarked text and decoding accuracy. Specifically, compared with MajorMark, MajorMark$^+$ (1) guarantees a larger expected $\gamma$ through a *block-wise majority bit-aware encoding* strategy; and (2) enhances decoding accuracy by replacing the boundary value-sensitive clustering-based decoding with a *deterministic decoding* method.

**Block-wise Majority Bit-aware Encoding.** As we discussed in Remark 1, when $b$ goes large, the improvement of $\gamma$ achieved by MajorMark becomes less significant as $\gamma$ converges to 0.5. Inspired by previous methods that divide the full message into several blocks (Yoo et al., 2024; Qu et al.,

2024; Jiang et al., 2025) to perform watermark encoding, MajorMark$^+$ is equipped with a *block-wise majority bit-aware encoding* method to slow the convergence of $\gamma$ given a finite $b$, especially when $b$ is large. We present an overview of MajorMark$^+$'s block-wise majority bit-aware encoding process at generation step $t$ in Figure 2b. Specifically, given any message $\mathbf{m} \in \{0,1\}^{b \geq 2}$, MajorMark$^+$ divides it into $r$ equal-sized blocks $\{\mathbf{m}_1, \mathbf{m}_2, \ldots, \mathbf{m}_r\}$. During the $t$-th generation step of the LLM $f$, MajorMark$^+$ pseudo-randomly assigns the $i$-th block $\mathbf{m}_i$ to the current token $x_t$, where $i = (\texttt{ID}(x_{t-2}) + \texttt{ID}(x_{t-1})) \mod r$, where $\texttt{ID}(\cdot)$ is the ID of a given token. As a result, each token contributes to encoding only a specific portion of the overall message.

To encode a given block $\mathbf{m}_i$ where $i \in [r]$, MajorMark$^+$ applies the same *majority bit-aware encoding* strategy used in MajorMark: the green list is constructed based on the block-specific majority bit $\lambda_i$, and the logits are perturbed accordingly. Importantly, the hash function of MajorMark$^+$ is in the form of $\texttt{Hash}(k, x_{t-1}, x_{t-2}, \lambda_i, h_{\lambda_i})$. Specifically, we further incorporate the occurrence of the majority bit $h_{\lambda_i}$ into the calculation of the hash seed $s$, which enables us to yield the deterministic decoding function of MajorMark$^+$, as discussed later.

We now present the formal guarantee on MajorMark$^+$'s expected $\gamma$ as follows in Theorem 2.

**Theorem 2** ($\gamma$ of MajorMark$^+$). *MajorMark$^+$ preserves the lower bound of the green list ratio as in MajorMark, i.e., $|\mathcal{G}| \geq 0.5|\mathcal{V}|$ and $\gamma \geq 0.5$. Moreover, for random $b$-bit binary messages, we have $\mathbb{E}_{\mathbf{m}}[\gamma] \approx 0.5 + 1/\sqrt{2\pi(b/r)}$.*

*Proof.* The detailed proof is in Appendix A.16. $\square$

**Remark 2.** *Given any finite $b$, MajorMark$^+$ always guarantees a larger expected $\gamma$ than MajorMark for any $r > 1$. This property brings an enhanced text quality for MajorMark$^+$. Recall from Remark 1 that MajorMark discards extreme cases where the message is entirely composed of zeros or ones, i.e., $\mathbf{m} = \mathbf{0}^b$ or $\mathbf{m} = \mathbf{1}^b$. MajorMark$^+$ similarly excludes such extreme messages at the block level. These infeasible cases are extremely rare and do not meaningfully affect the practical effectiveness of MajorMark$^+$ when $b$ is large. Specifically, the number of infeasible codes is given by $2^b - (2^{b/r} - 2)^r$. For instance, given a large $b = 32$ and $r = 2$, the total codes are $2^{32}$, while the number of infeasible codes is only $262{,}140$, accounting for approximately $0.006\%$ of the code space.*

**Deterministic Decoding.** When the message length $b$ is large and the watermarking bias $\delta$ is not large enough, the resulting shard-wise token occurrence count becomes less skewed. This weak signal may degrade the effectiveness of MajorMark's decoding process in specific cases, as clustering algorithms are unsupervised and inherently sensitive to data points near decision boundaries, failing to accurately separate the shards. To address this limitation, MajorMark$^+$ introduces a deterministic decoding method designed to recover the majority bit $\lambda_i$ and its occurrences $h_{\lambda_i}$ in the message for each block with high reliability. Specifically, given the hash function $\texttt{Hash}(k, x_{t-1}, x_{t-2}, \lambda_i, h_{\lambda_i})$, MajorMark$^+$ exhaustively enumerates all possible combinations of $\lambda_i$ and $h_{\lambda_i}$. For each combination, it reconstructs the corresponding shard-wise token occurrence count. The correct configuration yields a distinctly skewed count, while incorrect configurations result in near-uniform counts. By identifying the configuration that induces the sharpest skew, MajorMark$^+$ reliably recovers the correct values of $\lambda_i$ and $h_{\lambda_i}$. The message bits corresponding to those shards with top-$h_{\lambda_i}$ token occurrences are then assigned the recovered bit $\lambda_i$, while the remaining bits are assigned $1 - \lambda_i$. The full message is subsequently reconstructed by assembling all blocks. This deterministic approach eliminates the reliance on clustering, thereby avoiding sensitivity to ambiguous token counts and improving decoding accuracy. The detailed encoding and decoding algorithms of MajorMark$^+$ and the implementation of the hash function are presented in Appendix A.14.

**Decoding Complexity Analysis.** MajorMark$^+$ needs to identify both the majority bit $\lambda_i$ and its occurrence $h_{\lambda_i}$ for each block. When $\lambda_i = 1$, there are $b/(2r)$ candidate values for $h_{\lambda_i}$, and $b/(2r) - 1$ values when $\lambda_i = 0$, resulting in $b/r - 1$ total configurations per block. Since there are $r$ such blocks, the total number of decoding configurations is $r \times (b/r - 1) = b - r$. Each configuration requires one pass over the unverified token sequence $\mathbf{x}$. While this incurs a small overhead compared to the original MajorMark, it remains highly efficient relative to existing methods, such as CycleShift (Fernandez et al., 2023) and DepthW (Li et al., 2024), which require $2^b \times$ enumerations. We further provide a empirical runtime analysis of our methods and representative baselines in Section 4.2.

Table 2: Comparison of methods across various message lengths and watermarking biases. The PPL of non-watermarked text is 3.81, serving as a lower bound for the PPL of all watermarked texts.

| Message Length | Method | $\delta = 2$ | | | $\delta = 4$ | | | $\delta = 6$ | | | Avg. BA↑ | Avg. PPL↓ |
|---|---|---|---|---|---|---|---|---|---|---|---|---|
| | | BA↑ | PPL↓ | Top-5↑ | BA↑ | PPL↓ | Top-5↑ | BA↑ | PPL↓ | Top-5↑ | | |
| $b = 8$ | CycleShift | **100.00** | 5.09 | 90.72 | **100.00** | 8.64 | 81.43 | **100.00** | 15.44 | 75.65 | **100.00** | 9.06 |
| | DepthW | 95.62 | 4.53 | 91.65 | **100.00** | 8.24 | 79.53 | **100.00** | 25.49 | 61.13 | 98.54 | 12.75 |
| | MPAC | 99.38 | 5.03 | 90.41 | **100.00** | 8.74 | 81.44 | **100.00** | 16.06 | 74.52 | 99.79 | 9.28 |
| | RSBH | **100.00** | 4.80 | 91.19 | **100.00** | 6.36 | 89.30 | **100.00** | 8.66 | 86.87 | **100.00** | 6.61 |
| | MajorMark | **100.00** | 4.50 | 92.19 | **100.00** | 5.87 | **90.65** | **100.00** | 7.81 | 89.23 | **100.00** | 6.06 |
| | MajorMark$^+$ | **100.00** | **4.43** | **92.43** | **100.00** | **5.75** | 90.36 | **100.00** | **6.97** | **90.06** | **100.00** | **5.72** |
| $b = 32$ | MPAC | 89.22 | 5.03 | 90.09 | 98.75 | 9.37 | 80.75 | **100.00** | 15.32 | 75.35 | 95.99 | 9.91 |
| | RSBH | 89.38 | 4.68 | 92.15 | 97.66 | 6.43 | 88.48 | 97.50 | 8.47 | 87.13 | 94.85 | 6.53 |
| | MajorMark | 96.74 | **4.43** | 91.90 | 99.06 | 6.32 | 89.28 | **100.00** | 8.36 | 88.09 | 98.60 | 6.37 |
| | MajorMark$^+$ | **97.81** | 4.49 | **92.42** | **100.00** | **6.05** | **90.17** | **100.00** | **7.90** | **88.49** | **99.27** | **6.15** |
| $b = 64$ | MPAC | 81.48 | 4.99 | 90.08 | 93.59 | 8.64 | 82.01 | 96.48 | 15.95 | 75.14 | 90.52 | 9.86 |
| | RSBH | 81.25 | 4.93 | 91.80 | 90.39 | 6.41 | 88.91 | 97.58 | 9.22 | 87.06 | 89.74 | 6.85 |
| | MajorMark | 83.59 | 4.78 | 91.46 | 93.83 | **6.11** | **89.34** | 98.67 | 8.82 | 87.27 | 92.03 | 6.57 |
| | MajorMark$^+$ | **86.95** | **4.74** | **91.83** | **97.81** | 6.21 | 89.30 | **99.69** | **7.93** | **88.43** | **94.82** | **6.29** |

## 4 EXPERIMENTS

### 4.1 EXPERIMENTAL SETTINGS

**General Settings.** By default, we consider a total of 20 users, corresponding to 20 randomly generated messages. Each user submits two prompts, and each prompt is used to generate a text of $T/2 = 250$ tokens, resulting in $T = 500$ tokens per user for decoding. We compare our proposed methods with four state-of-the-art methods, including DepthW (Li et al., 2024), CycleShift (Fernandez et al., 2023), MPAC (Yoo et al., 2024), and RSBH (Qu et al., 2024). For DepthW and CycleShift, we only evaluate the case of $b = 8$, as their decoding time grows exponentially and becomes impractical for larger message lengths. For MajorMark, we use KMeans (McQueen, 1967) clustering with $K = 2$ for decoding, and all other hyperparameters of KMeans are kept at their default settings. For MajorMark$^+$, the default number of blocks $r$ is set to 2.

**Datasets and Models.** Following prior work (Kirchenbauer et al., 2023; Yoo et al., 2024; Qu et al., 2024), we conduct experiments primarily on the text completion task using the C4 news dataset (Raffel et al., 2020). Additional results on the Essays (Schuhmann, 2023) and OpenGen (Krishna et al., 2023) datasets are reported in Appendix A.7, while results on other benchmark tasks, such as *story generation* and *text summarization*, are provided in Appendix A.8 and Appendix A.9, respectively. Unless otherwise stated, we adopt the state-of-the-art open-source LLaMA-2-7B model (Touvron et al., 2023). Further results on other widely used public LLMs are given in Appendix A.6.

**Metrics.** We evaluate the quality of generated text using perplexity (PPL) computed by a larger LLaMA-2-13B model. In addition, we report the Top-5 hit rate, which measures the proportion of generated tokens that appear among the top-5 logits of the original model output, indicating how often desirable tokens are still sampled. In addition, we report metrics for the semantic and lexical preservation of watermarked text in Appendix A.19, where the results show that our method maintains higher semantic and lexical quality than the other methods. For decoding accuracy, we adopt bit accuracy (BA), following prior works (Zhu et al., 2018; Luo et al., 2020; Yang et al., 2022; Yoo et al., 2024), defined as the proportion of correctly decoded bits relative to the ground-truth message. A preferred watermarking method should achieve a low PPL and a high Top-5 hit rate and BA, respectively.

### 4.2 EMPIRICAL RESULTS

**Main Results.** We conduct a comprehensive evaluation of representative multi-bit watermarking methods under varying message lengths ($b \in \{8, 32, 64\}$) and watermarking biases ($\delta \in \{2, 4, 6\}$). More results under varying message length $b$ are presented in Appendix A.5. Table 2 reports BA, PPL, and Top-5 hit ratio for each setting. When $b = 8$, all methods achieve high BA across all watermarking bias settings. In this low-bit regime, MajorMark and MajorMark$^+$ exhibit the best

text quality, reflected by their average PPLs of 6.06 and 5.72, which are the second-lowest and lowest among all methods.

As $b$ increases, all methods observe a decline in both PPL and BA due to the increased difficulty in encoding and decoding longer messages. Nonetheless, MajorMark and MajorMark$^+$ consistently maintain strong performance, outperforming MPAC and RSBH in both PPL and BA. For example, when $b = 64$, MajorMark and MajorMark$^+$ achieve PPLs of 6.57 and 6.29, which represent improvements of $+0.28$ and $+0.56$ over RSBH's PPL of 6.85. These improvements stem from the majority bit-aware encoding strategy, which enables a larger green list during encoding, thereby better preserving the text quality. Furthermore, MajorMark and MajorMark$^+$ achieve higher Top-5 hit rates across all settings, further demonstrating their ability to maintain text quality. In terms of BA, they achieve 92.03% and 94.82%, respectively, outperforming MPAC's 90.52% by $+1.50\%$ and $+4.30\%$. This is because both MajorMark and MajorMark$^+$ avoid relying on token frequency counting within the green list, as done in MPAC and RSBH. Instead, their clustering-based and deterministic decoding methods ensure high decoding accuracy even with a large green list.

**Results on Available Tokens $T$ for Decoding.** Figure 4 illustrates the BA of different watermarking methods under varying numbers of observed tokens $T \in \{400, 450, 500, 550, 600\}$, evaluated under watermarking strengh $\delta = 2$ (left) and $\delta = 4$ (right), with a fixed message length of $b = 64$. As expected, all methods exhibit improved BA as $T$ increases. Notably, under $\delta = 2$, both MajorMark and MajorMark$^+$ consistently outperform MPAC and RSBH across all token counts, demonstrating superior decoding effi-

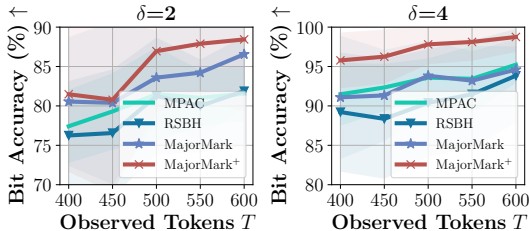

Figure 4: BA of different methods under varying numbers of observed tokens $T$ with $b = 64$.

ciency under constrained token availability. Under a larger watermarking bias $\delta = 4$, the advantage of MajorMark$^+$ becomes even more pronounced. For instance, at $T = 400$, MPAC and RSBH achieve BA scores of 89.22% and 91.48%, respectively, while MajorMark$^+$ achieves a significantly higher BA of 95.78%, yielding improvements of $+6.56\%$ over MPAC and $+4.30\%$ over RSBH. It is also worth noting that while MPAC achieves comparable BA to MajorMark under $\delta = 4$, this comes at the cost of reduced generation quality, as MPAC relies on a smaller green list ratio $\gamma$ to maintain robustness. For example, when $T = 500$, MPAC results in a PPL of 8.64, whereas MajorMark achieves a notably lower PPL of 6.11, yielding a quality improvement of $+2.53$.

**Robustness against Attacks.** We evaluate robustness against two widely studied post-generation text editing attacks: *Copy-Paste* and *Paraphrase* attacks (Zhang et al., 2024). Following previous works (Yoo et al., 2024; Qu et al., 2024), for the Copy-Paste attack, we randomly interleave the generated watermarked text into a non-watermarked text, mixing 10% of non-watermarked texts while maintaining the total length. For the Paraphrasing attack, we use the Dipper proposed in (Krishna et al., 2023) as the paraphraser.

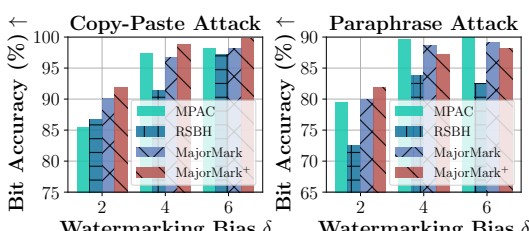

Figure 5: BA of different methods under Copy-Paste and Paraphrase attacks.

Figure 5 reports BA results under both attacks, evaluated across watermarking biases $\delta \in \{2, 4, 6\}$ with a fixed message length $b = 32$. For the Copy-Paste attack, MajorMark and MajorMark$^+$ generally outperform MPAC and RSBH across $\delta$ values, demonstrating their robustness even when some tokens are unwatermarked in the modified text. Paraphrase attacks remain a significant open challenge in the literature: all methods experience BA degradation under this stronger attack. While MPAC benefits from its small $\gamma$ and achieves higher BA at larger $\delta$, our methods maintain comparable performance to MPAC and significantly outperform RSBH. Notably, MajorMark achieves slightly higher BA than MajorMark$^+$, likely due to its simpler decoding process. More specifically, MajorMark$^+$ requires enumerating all combinations of majority bits and their occurrences for each message block. Under the Paraphrase attack, token occurrences become more diverse and less reliable. Additionally, we theoretically analyze the robustness of our methods under strong text

editing attacks in Appendix A.18. We also discuss the Spoofing (Jovanović et al., 2024) attack in Appendix A.4.

**Runtime Analysis.** We evaluate the computational overhead of our methods by measuring the average encoding and decoding time under our default settings. The empirical results are summarized in Table 3. For encoding, our methods add negligible overhead compared to standard generation ($\approx 36$s). For decoding, MajorMark performs two passes over the suspect text and applies a clustering step, giving a moderate overhead relative to MPAC (2.18s

Table 3: Comparison of runtime (in seconds) and BA. The decoding time for DepthW is estimated.

| Method | Encoding (s) | Decoding (s) | BA (%) |
|---|---|---|---|
| DepthW | 37.87 | $\approx 8.89 \times 10^8$ (Est.) | - |
| MPAC | 36.13 | 0.13 | 89.22 |
| RSBH | 36.06 | 25.56 | 89.38 |
| MajorMark | 36.58 | 2.18 | 96.74 |
| MajorMark$^+$ | 36.85 | 12.70 | 97.81 |

vs. 0.13s). MajorMark$^+$ requires $(b - r)$ decoding passes and reaches a total decoding time of 12.70s. This remains several orders of magnitude faster than exponential-time methods such as DepthW ($\approx 8.89 \times 10^8$s), and also faster than the baseline RSBH (25.56s). At the same time, our methods achieve much higher BA than existing approaches. This shows that our framework introduces only a modest computational cost while offering substantially improved robustness and reliability, making MajorMark and MajorMark$^+$ well-suited for real-world watermark verification.

**Ablation Study.** We conduct a comprehensive ablation study on two key design choices: the clustering method used by MajorMark and the number of message blocks $r$ in MajorMark$^+$. Table 4 summarizes the resulting BA and PPL with message length fixed at $b = 32$. For MajorMark, we evaluate two additional clustering methods beyond the default KMeans: Agglomerative Clustering (AC) (Müllner, 2011) and Gaussian Mixture Models (GMM) (Reynolds, 2015). With a strong watermarking bias ($\delta = 4$), all three methods achieve comparable BA.

Table 4: Ablation study of proposed MajorMark and MajorMark$^+$.

| Method | $\delta = 2$ | | $\delta = 4$ | | Avg. BA↑ | Avg. PPL↓ |
|---|---|---|---|---|---|---|
| | BA↑ | PPL↓ | BA↑ | PPL↓ | | |
| KMeans | 96.74 | 4.43 | 99.06 | 6.32 | 97.92 | 5.38 |
| AC | 94.06 | 4.43 | 99.84 | 6.32 | 96.95 | 5.38 |
| GMM | 92.81 | 4.43 | 99.84 | 6.32 | 96.33 | 5.38 |
| $r = 1$ | 95.94 | 4.74 | 100.00 | 6.55 | 97.97 | 5.65 |
| $r = 2$ | 97.81 | 4.49 | 100.00 | 6.05 | 98.91 | 5.27 |
| $r = 4$ | 90.00 | 4.42 | 98.44 | 5.61 | 94.22 | 5.02 |
| $r = 8$ | 91.88 | 4.46 | 98.59 | 5.20 | 95.24 | 4.83 |

When the bias is weaker ($\delta = 2$), KMeans slightly outperforms the others. Overall, the performance remains stable across different clustering choices, indicating that MajorMark is robust to the selection of the clustering method. For MajorMark$^+$, we vary the number of blocks $r \in \{1, 2, 4, 8\}$, where $r = 1$ denotes the case where the entire message is treated as a single block. We observe that increasing $r$ consistently improves PPL, supporting our theoretical insight on how block granularity influences the expected green list size $\gamma$. Notably, setting $r = 2$ favorably balances between BA and PPL, making it a practical configuration for real-world deployment of MajorMark$^+$.

**Additional Results and Discussions.** Examples of texts generated by different watermarking methods are provided in Appendix A.10. We further discuss how our method addresses practical false positive cases in Appendix A.11. Finally, directions for future work are presented in Appendix A.12.

## 5 CONCLUSION

We find that existing watermarking methods rely on counting how frequently generated tokens fall within the reconstructed green list to decode the embedded message. This approach inherently constrains the size of the green list to ensure decoding accuracy, but degrades the text quality. To address this, we propose MajorMark and MajorMark$^+$, both of which employ a novel majority bit-aware encoding strategy. This design enables the construction of larger green lists, as we theoretically prove. Furthermore, both methods are equipped with decoding algorithms that eliminate the need for green list token counting. Extensive experiments demonstrate that MajorMark and MajorMark$^+$ consistently outperform existing baselines in both text utility and decoding accuracy.

## ETHICS STATEMENT

This work strictly adheres to the ICLR Code of Ethics.

## REPRODUCIBILITY STATEMENT

We have made every effort to ensure the reproducibility of our results. The implementation of our methods is available in an anonymous GitHub repository (`https://anonymous.4open.science/r/MajorMark`). After acceptance, we will publicly release the full source code and detailed instructions to reproduce all experiments.

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

Table 5: Notation Table

| Symbol | Description |
|--------|-------------|
| $\gamma$ | The green list ratio |
| $\mathbf{m}$ | The binary multi-bit message |
| $m_i$ | The $i$-th bit of the message $\mathbf{m}$ |
| $b$ | The length of message $\mathbf{m}$ |
| $f$ | A large language model |
| $\mathbf{x}_p$ | Input prompt |
| $\mathbf{x}$ | The unverified text |
| $x_t$ | The $t$-th token generated by $f$ |
| $\mathcal{V}$ | The vocabulary of $f$ |
| $T$ | Total generation step of $f$ |
| $\mathbf{x}_g$ | The generated text of $f$ without watermark |
| $\mathbf{x}'_g$ | The generated text of $f$ with watermark |
| $T$ | The length of $\mathbf{x}_g$ |
| $\mathcal{G}$ | The green list |
| $\mathcal{R}$ | The red list |
| $\delta$ | The watermarking bias |
| $\text{Enc}(\cdot)$ | The encoding function for a watermarking method |
| $\text{Dec}(\cdot)$ | The decoding function for a watermarking method |
| $\mathbf{m}'$ | The decoded message via $\text{Dec}(\cdot)$ |
| $\text{Quality}(\cdot)$ | Quality function |
| $\text{PPL}(\cdot)$ | Conditional perplexity |
| $\text{ID}(\cdot)$ | The ID of a given token |
| $\tau$ | The tolerance margin on text quality for $\text{Enc}(\cdot)$ |
| $\lambda$ | Majority bit over a message $\mathbf{m}$ |
| $k$ | The secret hash key |
| $\text{Hash}(\cdot)$ | Hash function |
| $s$ | Pseudo-random seed generated by $\text{Hash}(\cdot)$ |
| $\mathcal{C}$ | Cluster with larger mean produced by MajorMark |
| $r$ | The number of blocks in MajorMark$^+$ |
| $h_0$ or $h_1$ | The occurrences of bit 0 or 1 in a message |
| $d$ | The number of bits in a message block |

# A APPENDIX

## A.1 NOTATION TABLE

We present the detailed notation table in Table 5.

## A.2 USE OF LLM STATEMENT

We used LLM solely for grammar checking and polishing the writing of this manuscript.

## A.3 HARDWARE SETTINGS

All experiments were carried out on a self-managed Linux-based computing cluster running Ubuntu 20.04.6 LTS. The cluster is equipped with eight NVIDIA RTX A6000 GPUs (each with 49 GB of memory) and AMD EPYC 7763 CPUs featuring 64 cores. Model inference leveraged GPU acceleration extensively. In total, the experiments accumulated roughly two weeks of GPU compute time.

## A.4 DISCUSSION ON SPOOFING ATTACK

**Discussion on Spoofing Attack.** Watermark spoofing, introduced by Jovanović et al. (2024), attempts to forge text that is falsely detected as watermarked, potentially harming the LLM provider's

Table 6: Performance of MajorMark and MajorMark$^+$ on Qwen2.5-7B, Gemma-2B, and LLaMA-3.1-8B with $b \in \{32, 64\}$ and $\delta \in \{2, 4\}$. The PPL for non-watermarked texts generated by Qwen2.5-7B, Gemma-2B, and LLaMA-3.1-8B are $4.97$, $5.71$, and $4.27$, respectively.

| Message Length | Model | Method | $\delta = 2$ | | | $\delta = 4$ | | | Avg. BA↑ | Avg. PPL↓ | Avg. Top-5↑ |
|---|---|---|---|---|---|---|---|---|---|---|---|
| | | | BA↑ | PPL↓ | Top-5↑ | BA↑ | PPL↓ | Top-5↑ | | | |
| $b = 32$ | Qwen2.5-7B | MajorMark | 96.25 | 6.69 | 86.99 | 99.22 | 8.49 | 85.29 | 97.74 | 7.59 | 86.14 |
| | | MajorMark$^+$ | 98.75 | 6.31 | 88.39 | 100.00 | 8.85 | 84.77 | **99.38** | **7.58** | **86.58** |
| | Gemma-2B | MajorMark | 96.41 | 7.02 | 87.16 | 99.06 | 9.80 | 83.38 | 97.74 | 8.41 | **85.27** |
| | | MajorMark$^+$ | 99.38 | 6.95 | 86.02 | 100.00 | 9.69 | 84.36 | **99.69** | **8.32** | 85.19 |
| | LLaMA-3.1-8B | MajorMark | 94.84 | 5.69 | 90.85 | 99.69 | 7.94 | 88.56 | 97.27 | 6.82 | 89.71 |
| | | MajorMark$^+$ | 97.81 | 5.78 | 90.90 | 100.00 | 7.57 | 88.75 | **98.91** | **6.68** | **89.83** |
| $b = 64$ | Qwen2.5-7B | MajorMark | 88.83 | 6.43 | 88.02 | 96.72 | 9.19 | 84.27 | 92.78 | 7.81 | 86.14 |
| | | MajorMark$^+$ | 92.03 | 6.31 | 88.20 | 99.22 | 9.00 | 84.25 | **95.63** | **7.66** | **86.23** |
| | Gemma-2B | MajorMark | 88.44 | 7.54 | 85.82 | 96.72 | 10.79 | 82.72 | 92.58 | 9.17 | 84.27 |
| | | MajorMark$^+$ | 93.75 | 7.41 | 85.93 | 99.38 | 9.82 | 82.75 | **96.56** | **8.62** | **84.34** |
| | LLaMA-3.1-8B | MajorMark | 86.17 | 5.33 | 92.11 | 95.94 | 7.58 | 88.48 | 91.06 | **6.46** | **90.30** |
| | | MajorMark$^+$ | 92.50 | 5.72 | 91.32 | 99.06 | 7.67 | 88.51 | **95.78** | 6.70 | 89.92 |

reputation. However, as noted in (Jiang et al., 2025), multi-bit watermarking embeds diverse message signals across tokens, which significantly increases the complexity of producing successful spoofs. Furthermore, Qu et al. (2024) show that when the watermarking method employs a hash function modeled as a random oracle (ROM) (Bellare & Rogaway, 1993), spoofing attacks become ineffective. Consequently, we do not consider spoofing attacks in this work.

## A.5 ADDITIONAL RESULTS ON MESSAGE LENGTH

We further investigate how the message length ($b \in \{32, 40, 48, 56, 64\}$) affects the performance of different watermarking methods, as shown in Figure 6. A larger $b$ increases the complexity of the decoding process, which generally leads to reduced BA. As expected, all methods exhibit a decline in BA as $b$ increases. MajorMark maintains a comparable BA to MPAC but achieves significantly better PPL. Notably, our proposed method, MajorMark$^+$, consistently yields the highest BA while maintaining a low PPL across all message lengths, demonstrating its robustness even when $b$ is as large as $64$.

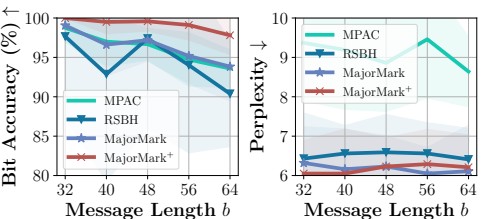

Figure 6: BA and PPL of different methods with increasing message length $b \in \{32, 40, 48, 56, 64\}$ with $\delta = 4$.

## A.6 RESULTS ON OTHER LLMs

We additionally report the performance of our methods on three other popular language models: Qwen2.5-7B (Yang et al., 2024), Gemma-2B (Team et al., 2024), and LLaMA-3.1-8B (Grattafiori et al., 2024). Specifically, we present the BA, PPL, and Top-5 hit rate results under message lengths $b \in \{32, 64\}$ and watermarking biases $\delta \in \{2, 4\}$ in Table 6. Note that for evaluating PPL, we use the larger models Qwen2.5-32B, Gemma-7B, and LLaMA-3.1-70B. As shown, both MajorMark and MajorMark$^+$ perform well across the board. These results demonstrate the strong generalization ability of MajorMark and MajorMark$^+$ across different LLM architectures and capacities.

## A.7 ADDITIONAL RESULTS ON OPENGEN AND ESSAYS DATASETS

We conduct a comprehensive evaluation of MPAC, RSBH, and our two proposed methods, MajorMark and MajorMark$^+$, on the OpenGen (Krishna et al., 2023) and Essays (Schuhmann, 2023) datasets. Results for message lengths $b \in \{32, 64\}$ and watermarking biases $\delta \in \{2, 4, 6\}$ are reported in Table 7 and Table 8, respectively.

Table 7: Performance comparison of watermarking methods (MPAC, RSBH, MajorMark, and MajorMark[+]) on the OpenGen dataset with message lengths $b \in \{32, 64\}$ and watermarking biases $\delta \in \{2, 4, 6\}$.

| Message Length | Method | $\delta = 2$ | | | $\delta = 4$ | | | $\delta = 6$ | | | Avg. BA↑ | Avg. PPL↓ | Avg. Top-5↑ |
|---|---|---|---|---|---|---|---|---|---|---|---|---|---|
| | | BA↑ | PPL↓ | Top-5↑ | BA↑ | PPL↓ | Top-5↑ | BA↑ | PPL↓ | Top-5↑ | | | |
| $b = 32$ | MPAC | 86.56 | 4.77 | 91.26 | 97.34 | 9.23 | 81.86 | 99.38 | 15.08 | 76.41 | 94.43 | 9.69 | 83.18 |
| | RSBH | 84.53 | 4.68 | 92.47 | 96.56 | 6.43 | 89.98 | 97.66 | 8.64 | 88.47 | 92.92 | 6.58 | 90.31 |
| | MajorMark | 90.94 | 4.35 | 92.83 | 97.81 | 6.02 | 90.01 | 99.38 | 8.51 | 87.95 | 96.04 | 6.29 | 90.26 |
| | MajorMark[+] | **92.03** | **4.09** | **93.85** | **99.38** | **5.63** | **91.16** | **100.00** | **7.96** | **88.90** | **97.14** | **5.89** | **91.30** |
| $b = 64$ | MPAC | 78.12 | 4.83 | 90.64 | 90.70 | 8.95 | 82.52 | 96.33 | 15.79 | 75.50 | 88.38 | 9.86 | 82.89 |
| | RSBH | 77.11 | 4.53 | **93.05** | 89.06 | 6.33 | **90.25** | 96.95 | 8.59 | 87.79 | 87.71 | 6.48 | 90.36 |
| | MajorMark | **84.06** | 4.77 | 92.24 | 93.44 | **6.14** | 89.97 | 96.02 | 8.42 | 88.22 | 91.17 | 6.44 | 90.14 |
| | MajorMark[+] | 79.92 | **4.41** | 92.62 | **97.66** | 6.37 | 89.73 | **99.06** | **8.01** | **88.98** | **92.21** | **6.26** | **90.44** |

Table 8: Performance comparison of watermarking methods (MPAC, RSBH, MajorMark, and MajorMark[+]) on the Essays dataset with message lengths $b \in \{32, 64\}$ and watermarking biases $\delta \in \{2, 4, 6\}$.

| Message Length | Method | $\delta = 2$ | | | $\delta = 4$ | | | $\delta = 6$ | | | Avg. BA↑ | Avg. PPL↓ | Avg. Top-5↑ |
|---|---|---|---|---|---|---|---|---|---|---|---|---|---|
| | | BA↑ | PPL↓ | Top-5↑ | BA↑ | PPL↓ | Top-5↑ | BA↑ | PPL↓ | Top-5↑ | | | |
| $b = 32$ | MPAC | 88.44 | 4.29 | 92.18 | 98.75 | 9.28 | 80.54 | 99.69 | 14.91 | 75.30 | 95.63 | 9.49 | 82.67 |
| | RSBH | 89.22 | 4.23 | 93.31 | 97.66 | 6.33 | 88.98 | 97.66 | 7.78 | 87.78 | 94.85 | 6.11 | 90.02 |
| | MajorMark | 91.41 | **4.12** | 93.24 | 98.28 | 5.98 | 89.60 | 99.53 | 7.71 | 87.70 | 96.41 | 5.94 | 90.18 |
| | MajorMark[+] | **96.72** | 4.21 | **93.37** | **100.00** | **5.83** | **90.44** | **100.00** | **7.40** | **88.92** | **98.91** | **5.81** | **90.91** |
| $b = 64$ | MPAC | 78.36 | 4.33 | 92.85 | 92.58 | 9.12 | 80.94 | 96.72 | 14.23 | 75.83 | 89.22 | 9.23 | 83.21 |
| | RSBH | 76.72 | **4.05** | 93.50 | 93.67 | 6.11 | 89.25 | 98.67 | 8.51 | 87.16 | 89.69 | 6.22 | 89.97 |
| | MajorMark | 82.73 | 4.33 | 92.56 | 94.84 | 6.07 | 89.48 | 96.88 | 8.46 | 87.07 | 91.48 | 6.29 | 89.70 |
| | MajorMark[+] | **87.19** | 4.14 | **93.56** | **98.75** | **5.99** | **89.83** | **99.69** | **7.95** | **87.42** | **95.21** | **6.03** | **90.27** |

Consistent with the trends observed on the C4 dataset, our methods consistently outperform the state-of-the-art baselines. In particular, MajorMark[+] achieves the highest average BA and Top-5 hit rate, while maintaining the lowest PPL across all settings. This demonstrates its strong ability to preserve the utility of watermarked text while ensuring high decoding accuracy. For example, on the Essays dataset with $b = 64$, MajorMark[+] achieves a BA of 95.21%, which is $+5.99\%$ and $+5.52\%$ higher than MPAC and RSBH, respectively. In terms of PPL, MajorMark[+] obtains the lowest score of 6.03, outperforming MPAC and RSBH by margins of 3.20 and 0.19, respectively. These results further confirm the strong generalization ability of both MajorMark and MajorMark[+] across diverse datasets, highlighting the practical effectiveness of our methods in varying domains.

## A.8 RESULTS ON STORY GENERATION TASK.

In this section, we evaluate existing methods alongside our two proposed approaches on the story generation task. Specifically, we use the WritingPrompts dataset (Fan et al., 2018) with the LLaMA-2-7B-chat model (Touvron et al., 2023). The chat prompt is designed to encourage coherent and imaginative story generation and is defined as follows:

> [System] *You are a helpful assistant that writes engaging and coherent stories.*
>
> [User] *Please write a detailed and imaginative short story based on the following prompt: [prompt]*

Results for message lengths $b \in \{32, 64\}$ and watermarking biases $\delta \in \{4, 6\}$ are summarized in Table 9. Our methods, MajorMark and MajorMark[+], consistently achieve strong performance on the story generation task. In particular, across both watermarking biases, MajorMark[+] yields the highest average BA and Top-5 hit ratios, as well as the lowest PPL, indicating its effectiveness in embedding reliable watermarks while preserving text quality. Although MajorMark performs slightly worse than MajorMark[+], it still outperforms existing state-of-the-art baselines, demonstrating its robustness and practicality.

Table 9: Performance comparison of watermarking methods (MajorMark and MajorMark$^+$) on the WritingPrompts dataset for the story generation task with message lengths $b \in \{32, 64\}$ and watermarking biases $\delta \in \{4, 6\}$.

| Message Length | Method | $\delta = 4$ | | | $\delta = 6$ | | | Avg. BA↑ | Avg. PPL↓ | Avg. Top-5↑ |
|---|---|---|---|---|---|---|---|---|---|---|
| | | BA↑ | PPL↓ | Top-5↑ | BA↑ | PPL↓ | Top-5↑ | | | |
| $b = 32$ | MPAC | 84.69 | 4.69 | 96.09 | 98.28 | 13.65 | 85.41 | 91.49 | 9.17 | 90.75 |
| | RSBH | 83.28 | 4.17 | 98.17 | 96.41 | 6.20 | 96.81 | 89.85 | 5.19 | 97.49 |
| | MajorMark | 89.38 | 4.09 | 98.70 | 96.41 | 6.19 | 97.12 | 92.90 | 5.14 | 97.91 |
| | MajorMark$^+$ | **91.88** | **3.91** | **98.72** | **98.44** | **5.52** | **97.51** | **95.16** | **4.72** | **98.12** |
| $b = 64$ | MPAC | 76.33 | 4.65 | 96.02 | 91.33 | 11.32 | 87.04 | 83.83 | 7.99 | 91.53 |
| | RSBH | 76.41 | 4.14 | 98.36 | 85.39 | 6.18 | 97.06 | 80.90 | 5.16 | 97.71 |
| | MajorMark | 80.94 | 4.05 | **98.93** | 93.05 | 6.08 | 97.13 | 87.00 | 5.07 | 98.03 |
| | MajorMark$^+$ | **83.83** | **4.00** | 98.69 | **94.53** | **5.63** | **97.40** | **89.18** | **4.82** | **98.05** |

Table 10: Performance comparison of watermarking methods (MajorMark and MajorMark$^+$) on the CNN/DailyMail dataset for the text summarization task with message lengths $b \in \{32, 64\}$ and watermarking biases $\delta \in \{4, 6\}$.

| Message Length | Method | $\delta = 4$ | | | $\delta = 6$ | | | Avg. BA↑ | Avg. PPL↓ | Avg. Top-5↑ |
|---|---|---|---|---|---|---|---|---|---|---|
| | | BA↑ | PPL↓ | Top-5↑ | BA↑ | PPL↓ | Top-5↑ | | | |
| $b = 32$ | MPAC | 83.12 | 6.36 | 96.68 | 96.41 | 12.25 | 90.48 | 89.77 | 9.31 | 93.58 |
| | RSBH | 85.00 | 5.81 | 98.83 | **97.34** | 8.07 | 97.24 | 91.17 | 6.94 | 98.04 |
| | MajorMark | 88.12 | 5.72 | 99.02 | 95.78 | 7.70 | 98.00 | 91.95 | 6.71 | 98.51 |
| | MajorMark$^+$ | **89.53** | **5.72** | **99.10** | 95.16 | **7.13** | **98.52** | **92.35** | **6.43** | **98.81** |
| $b = 64$ | MPAC | 78.67 | 6.55 | 97.06 | 91.33 | 12.86 | 90.09 | 85.00 | 9.71 | 93.58 |
| | RSBH | 73.05 | 5.99 | 98.81 | 84.92 | 7.87 | 97.31 | 78.99 | 6.93 | 98.06 |
| | MajorMark | 83.36 | 5.73 | 99.02 | 89.30 | 8.18 | 97.33 | 86.33 | 6.95 | 98.18 |
| | MajorMark$^+$ | **84.22** | **5.51** | **99.39** | **91.72** | **7.33** | **98.01** | **87.97** | **6.42** | **98.70** |

## A.9 RESULTS ON TEXT SUMMARIZATION TASK.

In this section, we evaluate existing methods alongside our two proposed approaches on the text summarization task. Specifically, we use the CNN/DailyMail dataset (Hermann et al., 2015) with the LLaMA-2-7B-chat model as the backbone. The chat prompt is designed to instruct the model to generate concise and faithful summaries given an input article, using the following format:

> [System] *You are a helpful assistant specialized in summarization. You take a document and write a concise, faithful summary.*
> [User] *Please summarize the following article in a few sentences: [article]*

Results for message lengths $b \in \{32, 64\}$ and watermarking biases $\delta \in \{4, 6\}$ are summarized in Table 10. Similar to the results observed for text completion and story generation, our methods consistently achieve both the highest text quality and the highest decoding accuracy across both watermarking bias settings. Together with the results reported in Appendix A.8, these findings demonstrate the strong generalization ability of our proposed methods, MajorMark and MajorMark$^+$, across diverse natural language processing tasks. This highlights their potential as practical candidates for real-world watermarking deployment.

## A.10 EXAMPLES OF GENERATED TEXTS

Table 12 presents examples of generated texts produced by different watermarking methods for a given prompt, with watermarking bias $\delta = 4$ and message length $b = 32$. As shown, both MajorMark and MajorMark$^+$ yield lower PPL, indicating better fluency in the generated text.

## A.11 DISCUSSION ON FALSE POSITIVE CASES

False positives are a common issue for all multi-bit watermarking methods. Even when given an unwatermarked text, the decoder will still output a message, which may be misinterpreted as a valid watermark. This issue is rarely discussed in prior work. Methods such as CycleShift, DepthW, and RSBH do not include any mechanism for detecting false positives. In contrast, our methods, MajorMark and MajorMark$^+$, naturally provide a way to identify unwatermarked text.

Table 11: Standard deviation statistics, TPR, and FPR for different message lengths $b$.

| $b$ | Watermarked ($\sigma_\lambda, \sigma_{1-\lambda}$) | Unwatermarked ($\sigma_1, \sigma_2$) | TPR / FPR |
|---|---|---|---|
| 32 | (8.68, 4.34) | (4.41, 4.57) | 100 / 5 ($\tau = 2$) |
| 24 | (10.55, 4.86) | (5.20, 5.43) | 95 / 0 ($\tau = 2$) |
| 16 | (16.22, 6.08) | (6.08, 6.34) | 100 / 0 ($\tau = 4$) |
| 8 | (28.26, 7.81) | (8.22, 8.60) | 100 / 0 ($\tau = 10$) |

Take MajorMark as an example. During decoding, the correct majority bit $\lambda$ is recovered by selecting the value that yields a more skewed token-count distribution. This step itself serves as an indicator for detecting false positives: if neither $\lambda = 0$ nor $\lambda = 1$ produces a sufficiently skewed distribution, the input is likely unwatermarked. In practice, one may set a threshold based on this skewness measure, or equivalently, the difference in standard deviations between the two hypotheses. If the difference falls below the threshold, the decoder concludes that the text is unwatermarked; otherwise, it proceeds with message recovery.

We further evaluated detection performance under $b = 32$ and $\delta = 2$ by comparing the standard deviations $\sigma_\lambda$ and $\sigma_{1-\lambda}$ across watermarked and unwatermarked texts. If their difference was below the threshold $\tau$, we labeled the text as unwatermarked; otherwise, we labeled it as watermarked. As shown in Table 11, the separability between the two groups remains clear across different message lengths $b$, even though the absolute values of the deviations change with $b$. These results show that our methods can effectively detect false positives, addressing a problem that existing multi-bit watermarking approaches do not handle.

### A.12 Limitations and Future Directions

We now discuss the limitations of our work. While our majority bit-aware encoding automatically determines the green list size for each message, this design imposes constraints on the flexibility of $\gamma$ in certain scenarios. In the future, it would be desirable to develop methods that preserve the advantages of MajorMark and MajorMark$^+$, namely, their independence from green list token frequencies, while allowing for a more flexible specification of $\gamma$. In addition, MajorMark$^+$ requires additional computation time to decode messages from the generated text. Although it remains more efficient than several existing approaches, further improving its decoding efficiency remains a promising direction for future work. Co-designing in the green list ratio $\gamma$ and the watermarking bias $\delta$ is also a promising future direction.

### A.13 Algorithms of MajorMark

**Encoding.** We present the detailed encoding procedure of MajorMark in Algorithm 1. The hash function $\texttt{Hash}(\cdot)$ can be implemented flexibly; in our implementation, we use the formula $(k \times x_{t-1} \times x_{t-2} + \lambda \times 31) \bmod 2^{64}$, where $k = 15{,}485{,}863$. For the permutation step $\texttt{Permute}(\cdot)$, we adopt PyTorch's built-in $\texttt{torch.randperm}$ to generate a permutation of the vocabulary. The $\texttt{Partition}(\cdot)$ function then divides the permuted vocabulary $\mathcal{V}'$ into $b$ equal-sized shards in order.

**Decoding.** The decoding procedure of MajorMark is detailed in Algorithm 2. To ensure consistency, the hash function $\texttt{Hash}(\cdot)$ used during decoding must exactly match the one used in encoding. Additionally, we incorporate the false positive case detection module into Algorithm 2. The function $\texttt{std}(\cdot)$ computes the standard deviation of a numeric array, while $\texttt{argmax}(\cdot)$ returns the index of the maximum value in a list. We employ the $\texttt{KMeans}(\cdot)$ clustering algorithm from the scikit-learn library (Pedregosa et al., 2011), with the number of clusters set to 2 and all other parameters set to their default values. The returned cluster $\mathcal{C}$ is the one with the higher mean shard occurrence count. The final message is reconstructed by assigning the majority bit $\lambda$ to shards in $\mathcal{C}$ and the complement bit $1-\lambda$ to the remaining shards.

### A.14 Algorithms of MajorMark$^+$

**Encoding.** The encoding procedure of MajorMark$^+$ is detailed in Algorithm 3. We begin by dividing the full message $\mathbf{m} \in 0, 1^b$ into $r$ disjoint blocks using the function $\texttt{divide}(\cdot)$. For each block $\mathbf{m}_w$, the function $\texttt{get\_majority\_bit}(\cdot)$ returns both the majority bit $\lambda_w$ and its frequency $h_{\lambda_w}$.

---

**Algorithm 1:** The Encoding Function of MajorMark

**Input** : User prompt $\mathbf{x}_p$, Maximum length $T$, Message $\mathbf{m} \in \{0,1\}^b$, Secret key $k$, LLM $f$,
Vocabulary $\mathcal{V}$, Bias strength $\delta$
**Output:** Watermarked text $\mathbf{x}'_g$

1   $\mathbf{x}_{:-1} \leftarrow \mathbf{x}_p$
2   $\lambda \leftarrow \texttt{get\_majority\_bit}(\mathbf{m})$
3   **for** $t = 0$ **to** $T - 1$ **do**
4      $x_{t-1}, x_{t-2} \leftarrow \mathbf{x}_{t-1}, \mathbf{x}_{t-2}$
5      $s \leftarrow \texttt{Hash}(k, x_{t-1}, x_{t-2}, \lambda)$
6      $\mathcal{V}' \leftarrow \texttt{permute}(\mathcal{V}, s)$
7      $\text{shard}_1, \ldots, \text{shard}_b \leftarrow \texttt{partition}(\mathcal{V}', b)$
8      $\mathcal{G} \leftarrow \emptyset$
9      **for** $i = 1$ **to** $b$ **do**
10         **if** $m_i == \lambda$ **then**
11           Append $\text{shard}_i$ to $\mathcal{G}$
12         **end**
13      **end**
14      $\ell^t \leftarrow f(\mathbf{x}_{:t})$ ;                 // Get next-token logits
15      **for** $j = 0$ **to** $|\mathcal{V}| - 1$ **do**
16         **if** $j \in \mathcal{G}$ **then**
17           $\ell^t_j \leftarrow \ell^t_j + \delta$
18         **end**
19      **end**
20      Sample $x_t \sim \texttt{Softmax}(\ell^t)$
21      Append $x_t$ to $\mathbf{x}$
22   **end**
23   **Return** $\mathbf{x}_{0:T-1}$

---

The hash function $\texttt{Hash}(\cdot)$ can be implemented flexibly; in our implementation, we use the formula $(k \times x_{t-1} \times x_{t-2} + \lambda_p \times 31 + h_{\lambda_p} \times 97) \bmod 2^{64}$, where $k = 15{,}485{,}863$. The permutation and partition functions, denoted $\texttt{Permute}(\cdot)$ and $\texttt{Partition}(\cdot)$, respectively, are consistent with those used in the original MajorMark encoding.

**Decoding.** The decoding procedure of MajorMark$^+$ is outlined in Algorithm 4. Since the message $\mathbf{m}$ is partitioned into $r$ blocks, we decode the message block by block. For each block, we consider both possible values of the majority bit hypothesis $\lambda' \in 0, 1$, along with all feasible values for its frequency $h_{\lambda'}$. Specifically, when $\lambda' = 0$, there are $b/r/2 - 1$ valid values for $h_{\lambda'}$, and when $\lambda' = 1$, there are $b/r/2$ possible values. We store token occurrence counts in a 3D matrix $\texttt{occ}$ of shape $r \times 2 \times b/r$.

The hash function $\texttt{Hash}(\cdot)$ used during decoding must exactly match the one used in the encoding process of MajorMark$^+$. For each combination of $(\lambda', h_{\lambda'})$, we compute the standard deviation of shard-wise token occurrences. The configuration that yields the highest standard deviation is selected, and we recover the corresponding majority bit $\lambda_p$ and frequency $h_{\lambda_p}$ for each block $p \in [r]$. To reconstruct the message, we identify the top-$h_{\lambda_p}$ shards (using $\texttt{Topk}$) with the highest counts in each block. The majority bit $\lambda_p$ is assigned to these shards, and the remaining shards are assigned the complement bit $1 - \lambda_p$. Finally, the decoded message $\mathbf{m}'$ is obtained by concatenating all block-wise results.

---

**Algorithm 2:** The Decoding Function of MajorMark with False Positive Detection

**Input** : Secret key $k$, LLM $f$, Vocabulary $\mathcal{V}$, Unverified text $\mathbf{x}$ of length $T$, False positive threshold $\tau$

**Output:** Decoded message $\mathbf{m}'$

1  Initialize $\texttt{occ} \leftarrow$ zero matrix of shape $2 \times b$ ; // Shard-wise token counts for both $\lambda'$

2  **for** $\lambda' \in \{0, 1\}$ **do**

3     **for** $t = 3$ **to** $T$ **do**

4        $x_{t-1}, x_{t-2} \leftarrow \mathbf{x}_{t-1}, \mathbf{x}_{t-2}$

5        $s \leftarrow \texttt{Hash}(k, x_{t-1}, x_{t-2}, \lambda')$

6        $\mathcal{V}' \leftarrow \texttt{permute}(\mathcal{V}, s)$

7        $\text{shard}_1, \ldots, \text{shard}_b \leftarrow \texttt{partition}(\mathcal{V}', b)$

8        **for** $i = 1$ **to** $b$ **do**

9           **if** $x_t \in \text{shard}_i$ **then**

10              $\texttt{occ}[\lambda'][i] \leftarrow \texttt{occ}[\lambda'][i] + 1$

11              **break**

12           **end**

13        **end**

14     **end**

15     $\sigma_{\lambda'} \leftarrow \texttt{std}(\texttt{occ}[\lambda'])$

16  **end**

17  **if** $|\sigma_0 - \sigma_1| \leq \tau$ **then**

18     **Return** False

19  **end**

20  $\lambda \leftarrow \texttt{argmax}(\sigma_0, \sigma_1)$

21  Cluster $\texttt{occ}[\lambda]$ into 2 groups via $\texttt{KMeans}(\cdot, 2)$

22  Let $\mathcal{C}$ be the cluster with higher average count

23  **for** $j = 1$ **to** $b$ **do**

24     **if** $j \in \mathcal{C}$ **then**

25        $m_j \leftarrow \lambda$

26     **else**

27        $m_j \leftarrow 1 - \lambda$

28     **end**

29  **end**

30  **Return** $\mathbf{m}' = (m_1, m_2, \ldots, m_b)$

---

**Algorithm 3:** The Encoding Function of MajorMark$^+$

**Input** : User prompt $\mathbf{x}_p$, Maximum length $T$, Message $\mathbf{m} \in \{0,1\}^b$, Secret key $k$, LLM $f$, Vocabulary $\mathcal{V}$, Bias strength $\delta$, Number of blocks $r$

**Output:** Watermarked text $\mathbf{x}_g'$

**1** Initialize $\mathbf{x}_{:-1} \leftarrow \mathbf{x}_p$

**2** Divide $\mathbf{m}$ into $r$ blocks: $\mathbf{m}_1, \ldots, \mathbf{m}_r \leftarrow \texttt{divide}(\mathbf{m}, r)$

**3 for** $w = 1$ **to** $r$ **do**

**4** $\quad$ $\lambda_w, h_{\lambda_w} \leftarrow \texttt{get\_majority\_bit}(\mathbf{m}_w)$

**5 end**

**6 for** $t = 0$ **to** $T-1$ **do**

**7** $\quad$ $x_{t-1}, x_{t-2} \leftarrow \mathbf{x}_{t-1}, \mathbf{x}_{t-2}$

**8** $\quad$ $p \leftarrow (x_{t-1} + x_{t-2}) \bmod r$ ; $\hspace{3cm}$ // Select block index

**9** $\quad$ $s \leftarrow \texttt{Hash}(k, x_{t-1}, x_{t-2}, \lambda_p, h_{\lambda_p})$

**10** $\quad$ $\mathcal{V}' \leftarrow \texttt{permute}(\mathcal{V}, s)$

**11** $\quad$ $\text{shard}_1, \ldots, \text{shard}_{b/r} \leftarrow \texttt{partition}(\mathcal{V}', b/r)$

**12** $\quad$ $\mathcal{G} \leftarrow \emptyset$

**13** $\quad$ **for** $i = 1$ **to** $b/r$ **do**

**14** $\quad\quad$ **if** $m_i == \lambda_p$ **then**

**15** $\quad\quad\quad$ Append $\text{shard}_i$ to $\mathcal{G}$

**16** $\quad\quad$ **end**

**17** $\quad$ **end**

**18** $\quad$ $\ell^t \leftarrow f(\mathbf{x}_{:t})$ ; $\hspace{3cm}$ // Get next-token logits

**19** $\quad$ **for** $j = 0$ **to** $|\mathcal{V}| - 1$ **do**

**20** $\quad\quad$ **if** $j \in \mathcal{G}$ **then**

**21** $\quad\quad\quad$ $\ell_j^t \leftarrow \ell_j^t + \delta$

**22** $\quad\quad$ **end**

**23** $\quad$ **end**

**24** $\quad$ Sample $x_t \sim \texttt{Softmax}(\ell^t)$

**25** $\quad$ Append $x_t$ to $\mathbf{x}$

**26 end**

**27 Return** $\mathbf{x}_{0:T-1}$

---

**Algorithm 4:** The Decoding Function of MajorMark$^+$

---

**Input** : Secret key $k$, LLM $f$, Vocabulary $\mathcal{V}$, Unverified text $\mathbf{x}'_g$ of length $T$, Number of blocks $r$
**Output:** Decoded message $\mathbf{m}'$

1 Initialize occ $\leftarrow$ zero matrix of shape $r \times 2 \times (b/r)$
2 $\mathcal{H}_0 \leftarrow \{1, \ldots, b/r/2 - 1\}$
3 $\mathcal{H}_1 \leftarrow \{1, \ldots, b/r/2\}$
4 **for** $\lambda' \in \{0, 1\}$ **do**
5    **for** $h' \in \mathcal{H}_{\lambda'}$ **do**
6      **for** $t = 3$ **to** $T$ **do**
7        $x_{t-1}, x_{t-2} \leftarrow \mathbf{x}_{t-1}, \mathbf{x}_{t-2}$
8        $s \leftarrow \texttt{Hash}(k, x_{t-1}, x_{t-2}, \lambda', h')$
9        $\mathcal{V}' \leftarrow \texttt{permute}(\mathcal{V}, s)$
10        shard$_1, \ldots,$ shard$_{b/r} \leftarrow \texttt{partition}(\mathcal{V}', b/r)$
11        $p \leftarrow (x_{t-1} + x_{t-2}) \bmod r$
12        **for** $i = 1$ **to** $b/r$ **do**
13          **if** $x_t \in$ shard$_i$ **then**
14            occ$[p][\lambda'][i] \leftarrow$ occ$[p][\lambda'][i] + 1$
15            **break**
16          **end**
17        **end**
18      **end**
19    **end**
20 **end**
21 **for** $p = 0$ **to** $r - 1$ **do**
22    Initialize best_std $\leftarrow -1$, $(\lambda_p, h_{\lambda_p}) \leftarrow (0, 0)$
23    **for** $\lambda' \in \{0, 1\}$ **do**
24      **for** $h' \in \mathcal{H}_{\lambda'}$ **do**
25        $\sigma \leftarrow \texttt{std}($occ$[p][\lambda'][0 : b/r])$
26        **if** $\sigma >$ best_std **then**
27          best_std $\leftarrow \sigma$
28          $(\lambda_p, h_{\lambda_p}) \leftarrow (\lambda', h')$
29        **end**
30      **end**
31    **end**
32 **end**
33 Initialize $\mathbf{m}' \leftarrow []$
34 **for** $p = 0$ **to** $r - 1$ **do**
35    $c \leftarrow$ occ$[p][\lambda_p]$
36    $\mathcal{T}_p \leftarrow \texttt{Topk}(c, h_{\lambda_p})$
37    **for** $j = 0$ **to** $b/r - 1$ **do**
38      **if** $j \in \mathcal{T}_p$ **then**
39        $m_{p,j} \leftarrow \lambda_p$
40      **end**
41      **else**
42        $m_{p,j} \leftarrow 1 - \lambda_p$
43      **end**
44    **end**
45    Append $(m_{p,0}, \ldots, m_{p,b/r-1})$ to $\mathbf{m}'$
46 **end**
47 **Return** $\mathbf{m}'$

---

Table 12: Generated texts under different methods.

**Prompt**: *Sports Academy has found "The Mamba Mentality." The 96,000-square-foot training facility in Thousand Oaks announced a partnership with Kobe Bryant. The 96,000-square-foot holistic athletic training facility in Thousand Oaks announced a partnership with former Lakers superstar Kobe Bryant on Monday morning. [...]*

| Method | Generated Text | PPL↓ |
| --- | --- | --- |
| No Watermark | short film. Bryan, who is a co-owner of the MLS team Los Angeles Football Club, has said he wants to open a youth soccer academy. He is also the co-owner of Granite Hill Capital Partners, a venture capital firm. His investment firm Easy Street has invested in more than 150 startups, according to Forbes. Sports Academy is planning to open a new location in the Valley in 2020. It's unclear if the new facility will be rebranded as Mamba Sports Academy. Sports reporter Nathaniel Percy contributed to this report. This article is written by Nathaniel Percy from Ventura County Star, Thousand Oakes and was legally licensed via the Tribune Content Agency through the NewsCred publisher network. Please direct all licensing questions to legal@newscred.com. © Copyright 2019 Ventura County Star. All rights reserved. This material may not be published, broadcast, rewritten or redistributed. Ventura County Star is the leading media company in Ventura County, California. The | 3.01 |
| MPAC | short. This article is written by Mike Coppinger from Los Angeles Daily News and was legally licensed via the Tribune Content Agency through the NewsCred publisher network. Please direct all licensing questions to [email protected] . Next Post: The top 265 hotel openings, renovations set to launch in 2023 This article is really interesting. Thanks for the great post! Governors Island is opening a sports arena with artificial turf and more. Based in a city that boasts an impressive number of parks and recreation centers, Washington DC still... Tennis courts are among the popular sports amenities in resorts. Tennis courts have long been staple sports amenities in vacation homes, luxury hotels, and resorts. Today's... Leading Resort Operators Launch New Brand Campaign, 'Live the High Life' Leading resort operator Great Western Resorts has united with its parent, Highgate Hotels, as well as... Tennis court construction is easy if you are properly prepared. Tennis court surfaces come in a variety of different | 8.58 |
| RSBH | He'll host a workout and clinic at the academy this spring and will be involved at the academy through social media promotions. He's already been involved through social media promos. "As a professional athlete who has spent nearly two decades training at world-class facilities, I know the value of sports academies and the importance they have in the lives of youth," Bryant said in a statement. "Sports Academy is a state of the art, all-inclusive facility that offers unparalleled opportunities to develop the body and the mind. "My team and I are thrilled to have our brand aligned alongside the most progressive training facility in the world and the entire MAMBA team," said Faulkner in a statement. "...We look forward to this partnership, in which we can share the same values and principles with each other." Want to see more from our team at the Ventura County Star? You must be a digital subscriber to view this content. You must be registered with your Ventura County Star account to view this content! You must be signed in to your Ventura County account to | 6.09 |
| MajorMark | film for "Dear Basketball." He is also an executive producer on the "What Doesn't Kill You" Netflix series. Sports Academy also operates training facilities in Beverly Hills, Santa Barbara and Newport Beach. A fifth center is set to open in Westlake Village in early February 2020. Sports Leagues and Activities Conejo Hills Elementary School in Thousand... Conejo High School District adopts new... Thousand Oaks school district reinstates mask mandate Calabasas school district to reinstate mask mandate Firefighters battle blaze in Simi Valley; evacuations ordered in Thousand Palms Thousand Palms fire update: 5,563 acres, 10Thousand palms wildfire: 5,563 acres, no structures lost, evacuations lifted Thousand Hills Fire: Cause of wildfire under investigation; mandatory evacuations lifted for most residents Thousand-Hills Fire: Cause under investigation; mandated evacuations lifted but evacuations | 5.74 |
| MajorMark$^+$ | video. This article is written by Samantha Mascarenas from The Ventura County Star and was legally licensed via the Tribune Content Agency through the NewsCred publisher network. Please direct all licensing questions to [email protected]. Copyright 2019, Ventura County Star. All rights reserved. From The Los Angeles Daily News, California. Distributed by Tribune Content Agency, LLC. Want to be the first to hear what's new in the fitness space and in our industry? Subscribe to the Sweat Equity Podcast! ©Copyright 2107. All rights Reserved. FitLife Brands, Inc. "The Sweat Equality Podcast" and "Sweat Equality" is the property and trademark of FitLife Brends,Inc. All rights Resereved. You can't do it alone, let us help you. We can do the heavy lifting for you, so you'll have more time to focus on your business. Tell us what you need and we'll take care of it. You're not alone. | 4.47 |

## A.15 Proof of the Guaranteed Green List Size of MajorMark

**Assumption.** We begin with an assumption on the distribution of bits in the embedded message.

**Assumption 1** (Uniform Bit Distribution). *Each bit in the message $\mathbf{m} \in \{0,1\}^b$ is sampled independently and uniformly at random. That is, for all $i \in \{1, \ldots, b\}$,*

$$\Pr(m_i = 0) = \Pr(m_i = 1) = 0.5, \text{ and } m_i \perp m_j \text{ for } i \neq j.$$

**Preliminaries.** We will make use of two theoretical tools: the De Moivre–Laplace Theorem for approximating the binomial distribution and a result on the mean absolute deviation of a normal distribution.

**Theorem 3** (De Moivre–Laplace Central Limit Theorem). *Let $X_n \sim \text{Binomial}(n, p)$. Then for any real numbers $a < b$, the following convergence holds:*

$$\lim_{n \to \infty} \Pr\left(a \leq \frac{X_n - np}{\sqrt{np(1-p)}} \leq b\right) = \int_a^b \frac{1}{\sqrt{2\pi}} e^{-z^2/2} \, dz.$$

*Equivalently, the standardized variable*

$$Z_n := \frac{X_n - np}{\sqrt{np(1-p)}}$$

*converges in distribution to a standard normal variable:*

$$Z_n \xrightarrow{d} \mathcal{N}(0,1), \quad \text{as } n \to \infty.$$

*As a consequence, for large $n$, the binomial distribution can be approximated as:*

$$X_n \approx \mathcal{N}(np, \, np(1-p)).$$

**Proposition 1** (Mean Absolute Deviation of Normal Distribution). *Let $Z \sim \mathcal{N}(\mu, \sigma^2)$. Then the expected absolute deviation from the mean is given by:*

$$\mathbb{E}[\,|Z - \mu|\,] = \sigma\sqrt{\frac{2}{\pi}}.$$

*Proof.* We evaluate the expectation by integrating:

$$\mathbb{E}[\,|Z - \mu|\,] = \int_{-\infty}^{\infty} |z - \mu| \cdot \frac{1}{\sqrt{2\pi\sigma^2}} e^{-(z-\mu)^2/(2\sigma^2)} \, dz.$$

Applying the change of variable $x = \frac{z-\mu}{\sigma}$, we obtain:

$$\mathbb{E}[\,|Z - \mu|\,] = \sigma \int_{-\infty}^{\infty} |x| \cdot \frac{1}{\sqrt{2\pi}} e^{-x^2/2} \, dx = \sigma\sqrt{\frac{2}{\pi}},$$

which completes the proof. $\qquad \square$

**Main Proof.** We now prove the expected green list size under Assumption 1.

*Proof.* Let $\mathbf{m} = (m_1, m_2, \ldots, m_b) \in \{0,1\}^b$ be the embedded message. During watermark encoding, the vocabulary $\mathcal{V}$ is partitioned into $b$ equal-sized shards, each of size $|\mathcal{V}|/b$, with each shard corresponding to a bit $m_i$ in the message.

Let $h_0$ and $h_1$ denote the number of zeros and ones in $\mathbf{m}$, respectively. Then $h_0 + h_1 = b$, and under Assumption 1, we have $h_1 \sim \text{Binomial}(b, 0.5)$.

We define the majority bit $\lambda \in \{0, 1\}$ as:

$$\lambda = \begin{cases} 1 & \text{if } h_1 \geq h_0, \\ 0 & \text{otherwise.} \end{cases}$$

The green list $\mathcal{G}$ is constructed as the union of shards whose corresponding bits equal $\lambda$. Therefore, the size of the green list is given by:

$$|\mathcal{G}| = \max(h_0, h_1) \cdot \frac{|\mathcal{V}|}{b}.$$

Defining the green list ratio as $\gamma := \frac{|\mathcal{G}|}{|\mathcal{V}|}$, we observe that:

$$\gamma = \frac{\max(h_0, h_1)}{b} \geq \frac{1}{2},$$

since the maximum of two nonnegative numbers summing to $b$ is always at least $b/2$.

We now proceed to compute the expected value of $\gamma$, or equivalently, the expected size of $|\mathcal{G}|$. Taking expectation over all messages, we have:

$$\mathbb{E}_{\mathbf{m}}[|\mathcal{G}|] = \mathbb{E}\left[\max(h_1, b - h_1)\right] \cdot \frac{|\mathcal{V}|}{b}.$$

Noting that:

$$\max(h_1, b - h_1) = \frac{b}{2} + \left|h_1 - \frac{b}{2}\right|,$$

we obtain:

$$\mathbb{E}\left[\max(h_1, b - h_1)\right] = \frac{b}{2} + \mathbb{E}\left[\left|h_1 - \frac{b}{2}\right|\right].$$

By Theorem 3, for large $b$, the binomial variable $h_1 \sim \mathrm{Binomial}(b, 0.5)$ is approximated by a normal distribution:

$$h_1 \approx \mathcal{N}\left(\frac{b}{2}, \frac{b}{4}\right).$$

Then, by Proposition 1, the expected absolute deviation satisfies:

$$\mathbb{E}\left[\left|h_1 - \frac{b}{2}\right|\right] \approx \sqrt{\frac{b}{4}} \cdot \sqrt{\frac{2}{\pi}} = \sqrt{\frac{b}{2\pi}}.$$

Substituting into the previous expression yields:

$$\mathbb{E}_{\mathbf{m}}[|\mathcal{G}|] \approx \left(\frac{b}{2} + \sqrt{\frac{b}{2\pi}}\right) \cdot \frac{|\mathcal{V}|}{b} = \left(\frac{1}{2} + \frac{1}{\sqrt{2\pi b}}\right) |\mathcal{V}|.$$

Finally, we conclude that the expected green list ratio is:

$$\mathbb{E}_{\mathbf{m}}[\gamma] = \frac{\mathbb{E}_{\mathbf{m}}[|\mathcal{G}|]}{|\mathcal{V}|} \approx \left(\frac{1}{2} + \frac{1}{\sqrt{2\pi b}}\right),$$

which completes the proof. $\qquad\square$

### A.16 PROOF OF THE GUARANTEED GREEN LIST SIZE OF MAJORMARK$^+$

*Proof.* MajorMark$^+$ divides the message $\mathbf{m} \in \{0, 1\}^b$ into $r$ equal-sized blocks $\{\mathbf{m}_1, \ldots, \mathbf{m}_r\}$, where each block $\mathbf{m}_j$ has length $b/r$. For each block, a green list $\mathcal{G}_j$ is independently constructed over the full vocabulary $\mathcal{V}$ using the same *majority bit-aware encoding* rule as in MajorMark.

Let $\gamma_j := |\mathcal{G}_j|/|\mathcal{V}|$ denote the green list ratio for block $j$. As shown in the previous proof, for any message block of length $b/r$, the following lower bound always holds:

$$\gamma_j \geq \frac{1}{2}, \quad \text{for all } j \in \{1, \ldots, r\}.$$

We define the overall green list ratio $\gamma$ as:

$$\gamma := \frac{1}{r} \sum_{j=1}^{r} \gamma_j.$$

As a result, we have:

$$\gamma \geq \frac{1}{r} \sum_{j=1}^{r} \frac{1}{2} = \frac{1}{2},$$

which guarantees that MajorMark$^+$ always yields a green list containing at least half of the vocabulary.

We now proceed to analyze the expected value of $\gamma$. From Theorem 1, we know that for each block of length $b/r$, the expected green list ratio is:

$$\mathbb{E}_{\mathbf{m}_j}[\gamma_j] \approx \frac{1}{2} + \frac{1}{\sqrt{2\pi(b/r)}}, \quad \forall j.$$

Hence, the expected value of the overall green list ratio is:

$$\mathbb{E}_{\mathbf{m}}[\gamma] = \mathbb{E}_{\mathbf{m}}\left[\frac{1}{r} \sum_{j=1}^{r} \gamma_j\right] = \frac{1}{r} \sum_{j=1}^{r} \mathbb{E}_{\mathbf{m}_j}[\gamma_j]$$

$$\approx \left(\frac{1}{2} + \frac{1}{\sqrt{2\pi(b/r)}}\right).$$

Multiplying by $|\mathcal{V}|$, we also obtain the expected green list size:

$$\mathbb{E}_{\mathbf{m}}[|\mathcal{G}|] = \mathbb{E}_{\mathbf{m}}[\gamma] \cdot |\mathcal{V}| \approx \left(\frac{1}{2} + \frac{1}{\sqrt{2\pi(b/r)}}\right) |\mathcal{V}|,$$

which completes the proof. □

## A.17 THEORETICAL JUSTIFICATION OF CLUSTERING-BASED DECODING

Under the random oracle assumption (Bellare & Rogaway, 1993), shard selections behave as independent random draws. By the Central Limit Theorem (Feller, 1991), the observed count $f_i$ of each shard $i$ is well-approximated by a Gaussian distribution. Applying a positive watermark bias $\delta$ increases the sampling probability of green shards (encoding the majority bit $\lambda$) from $p_R$ to a larger value $p_G$. This produces two groups of shards with distinct means:

$$f_i \sim \begin{cases} \mathcal{N}(\mu_G, \sigma_G^2), & \text{if shard } i \text{ encodes } \lambda, \\ \mathcal{N}(\mu_R, \sigma_R^2), & \text{if shard } i \text{ encodes } 1 - \lambda, \end{cases}$$

where $\mu_G = T \cdot p_G$ and $\mu_R = T \cdot p_R$. This defines a bimodal Gaussian mixture over shard counts. Since `KMeans` is equivalent to a hard-assignment EM procedure for a Gaussian Mixture Model with isotropic covariance, it is naturally suited for recovering the two clusters. Our ablation study (Table 4) directly evaluates Gaussian Mixture Models as an alternative decoder. GMM achieves decoding accuracy comparable to `KMeans` (for example, 96.33% vs. 97.92%), confirming that the empirical distribution of shard counts aligns with the Gaussian mixture model. These observations support both the modeling assumption and the choice of clustering-based decoding.

**Error analysis via distribution overlap.** Decoding errors occur when the clustering algorithm misclassifies a green shard as red or vice versa. The error probability is determined by the overlap between the two Gaussian components $\mathcal{N}(\mu_G, \sigma_G^2)$ and $\mathcal{N}(\mu_R, \sigma_R^2)$.

*Insufficient text length.* When $T$ is small, the variance of each Gaussian component is large relative to the mean gap $|\mu_G - \mu_R|$, causing substantial overlap. This leads to higher misclassification rates and explains the drop in BA for short outputs, as shown in Figure 4.

*Weak watermark bias.* When $\delta$ is small, the sampling probabilities $p_G$ and $p_R$ become nearly equal. Even for large $T$, the two means collapse into a single mode, producing a unimodal distribution. In this regime, `KMeans` cannot recover meaningful cluster centers and behaves like random guessing. This matches our results in Table 2, where a smaller $\delta$ yields lower decoding accuracy.

These analyses provide a theoretical explanation for the robustness and limitations of our decoding design. We thank the reviewer for raising this point, and we have incorporated this justification into the revised manuscript.

## A.18 ANALYSIS OF ROBUSTNESS UNDER TEXT EDITING ATTACKS

This section provides a unified analysis of why both MajorMark and MajorMark$^+$ retain robustness under paraphrasing attacks, despite the strong semantic rewrites introduced by paraphrasers such as *Dipper*. The stability of the two methods follows from (1) the statistical nature of the noise created by paraphrasing and (2) the decoding procedures that remain effective as long as the underlying shard distributions preserve a detectable gap.

**Statistical behavior of paraphrasing noise.** A paraphraser replaces a subset of watermarked tokens with contextually similar alternatives. These replacements produce two effects on the shard-count distribution. *1. Surviving tokens.* Paraphrasing does not rewrite all positions. Tokens that remain unchanged continue to vote for their correct shards, preserving part of the watermarking signal. *2. Replaced tokens.* From the decoder's viewpoint, replaced tokens behave like random samples from the vocabulary. Due to the majority-bit encoding, the green-shard ratio is at least $0.5$. This ensures that replacements dilute the signal rather than destroying it. In imbalanced cases where $h_\lambda > b/2$, the replacements even introduce a slight bias toward the green component. *3. Overall effect.* The paraphraser weakens the signal but does not erase it. The shard-count distribution remains biased toward the majority-bit shards, maintaining a non-zero gap that decoding algorithms can recover.

**Robustness of clustering-based decoding in MajorMark.** Recall that in Appendix A.17, each shard count $f_i$ is approximated by a Gaussian. The watermark bias $\delta$ creates two groups of shards with means $\mu_G = T p_G$ and $\mu_R = T p_R$, with $p_G > p_R$. This produces a bimodal Gaussian mixture: $f_i \sim \text{Mixture}\left(\mathcal{N}(\mu_G, \sigma_G^2), \mathcal{N}(\mu_R, \sigma_R^2)\right)$. Paraphrasing reduces the gap $\Delta\mu = \mu_G - \mu_R$, but the mixture remains separable as long as the gap exceeds the noise scale. `KMeans`, which is equivalent to a hard-assignment EM procedure for a Gaussian Mixture Model with isotropic covariance, only requires bimodality rather than exact mean values. Even when the two modes move closer, `KMeans`

Table 13: Semantic and lexical fidelity (BERTScore ↑ / ROUGE-1 ↑ / ROUGE-Lsum ↑) for different bias values $\delta$ with $b = 32$.

| Method | $\delta = 2$ | $\delta = 4$ | $\delta = 6$ |
|---|---|---|---|
| MPAC | 0.8376/0.33/0.29 | 0.8210/0.28/0.25 | 0.8105/0.26/0.22 |
| RSBH | 0.8384/0.35/0.31 | 0.8316/0.33/0.29 | 0.8234/0.30/0.26 |
| MajorMark | 0.8415/0.37/0.33 | 0.8305/0.33/0.28 | 0.8269/0.31/0.28 |
| MajorMark$^+$ | 0.8449/0.36/0.32 | 0.8381/0.34/0.29 | 0.8311/0.33/0.29 |

identifies the two clusters correctly if they remain distinguishable. This explains why MajorMark achieves stable decoding under strong paraphrasing.

**Robustness of block-wise decoding in MajorMark$^+$.** MajorMark$^+$ extends MajorMark with a block-wise structure. Paraphrasing can modify the context used to compute the block index $i$, causing tokens to be assigned to the wrong blocks during decoding. This does not form a structural weakness for the following reasons. *1. Uniform noise across blocks.* When context tokens are altered, the resulting block index becomes random. The contribution of the replaced token is therefore spread uniformly across all blocks rather than targeted at a specific block. *2. Graceful degradation.* A wrongly assigned token does not vote against the correct block. It simply becomes noise. Because the green-shard ratio is high, the correctly synchronized tokens provide enough evidence for recovering the block message. *3. Independence across blocks.* Each block is decoded independently. Noise affecting one block does not propagate to the others. This prevents local errors from causing global decoding failure.

**Summary.** Both MajorMark and MajorMark$^+$ are robust to paraphrasing noise because the attacks dilute the shard distribution without removing its core structure. The decoding mechanisms rely on separability rather than exact alignment. As long as a detectable statistical gap remains, our methods recover the embedded message reliably.

## A.19 SEMANTIC AND LEXICAL FIDELITY OF WATERMARKED TEXTS

Recall that in evaluation, we mainly use PPL as the metric for evaluating the quality of watermarked texts. To provide a more complete analysis, we further evaluate BERTScore (semantic similarity) and ROUGE-1 / ROUGE-Lsum (lexical and structural overlap) between watermarked and unwatermarked outputs generated from the same prompts. Table 13 reports results under varying values of $\delta$ with $b = 32$. Across all settings, MajorMark$^+$ achieves the highest BERTScore, indicating stronger semantic preservation than MPAC and RSBH. In addition, the ROUGE metrics show that both MajorMark and MajorMark$^+$ maintain a high degree of lexical and structural similarity to the unwatermarked outputs. These results suggest that the large green list in our methods allows the model to retain its natural word choices and phrasing, rather than forcing substitutions that may degrade semantic quality.

## A.20 IMPACT OF MESSAGE IMBALANCE ON DECODING ACCURACY

This section analyzes how the distribution of bit values in the embedded message influences decoding behavior. When the message contains a highly uneven number of 0s and 1s, the number of green shards grows and the green list ratio $\gamma$ approaches 1.0. Under a fixed text length $T$, the probability boost introduced by the watermark bias $\delta$ is then distributed across many green shards. This reduces the expected count per green shard and decreases the mean gap between green and red shards, producing a weaker watermarking signal for decoding.

At the same time, this scenario introduces a beneficial property. When $\gamma$ becomes very large (for example, above 80%), the red list becomes a small fraction of the vocabulary. In this case, the watermark bias $\delta$ can be increased without degrading text quality, since most tokens remain unrestricted during generation. A larger $\delta$ amplifies the separation between the green and red shard distributions and compensates for the reduced per-shard signal caused by the imbalance. We evaluated this effect using MajorMark$^+$ with messages where one bit value occupies more than 80% of a $b = 32$ message. With a stronger bias of $\delta = 4$, MajorMark$^+$ achieves a BA of 97.19% and a PPL of 4.36. These results are comparable to those obtained under randomly generated messages with $\delta = 2$ (BA:

97.81%, PPL: 4.49). This confirms that increasing $\delta$ effectively offsets the weaker per-shard signal when the embedded message is imbalanced.

In conclusion, although imbalanced messages reduce the per-shard watermarking signal, the resulting increase in $\gamma$ allows the use of a larger bias $\delta$, which restores the separability between green and red shards. Both analysis and experiments show that message imbalance does not undermine the robustness of MajorMark or MajorMark$^+$.

