# OpenReview forum: "Majority Bit-aware Watermarking for Large Language Models"
_ICLR.cc/2026/Conference — Submitted to ICLR 2026_

### Official Review · Reviewer_tnCJ · 2025-10-26

**Soundness:** 2
**Presentation:** 3
**Contribution:** 2
**Rating:** 4
**Confidence:** 4

**Summary:**

The paper presents a multi-bit watermarking technique for large language models. The proposed approach selects tokens from a green list that covers the majority of the vocabulary, aiming to maintain the quality of generated text. Building on a majority-bit–aware watermarking framework, the authors introduce two methods: MajorMark and MajorMark+, with the latter offering a higher expected green-list ratio and better suitability for embedding longer messages. Experimental results demonstrate the effectiveness of the proposed watermarking scheme.

**Strengths:**

+ This is an interesting and important topic.
+ The paper is clearly written and easy to follow. The authors present two variants of their majority-bit–aware watermarking scheme, providing a structured and coherent discussion of their design and application.

**Weaknesses:**

- The paper claims that previous methods suffer from high computational complexity and that the proposed approaches are computationally efficient. However, the work lacks a detailed comparison or empirical analysis to substantiate this claim. In practice, the proposed methods appear to introduce additional computational costs due to multiple rounds of hashing, clustering-based decoding, and trial-and-error decoding processes.
- The decoding accuracy heavily depends on the specific embedded information. When the numbers of majority and minority bits are similar, decoding accuracy tends to be higher. In contrast, when the embedded bits are highly imbalanced (e.g., significantly more 1s than 0s), the variance in token distribution across shards diminishes, making decoding less effective. For instance, embedding 4 bits into 16 tokens with the message 1100 may yield shard distributions of {8, 8, 0, 0}, while 1110 could result in {5, 5, 6, 0}, illustrating that the decoding feature, variance in shard token counts, depends on the specific bit pattern.
- The proposed system cannot embed messages consisting entirely of 0s or 1s, or containing long substrings of identical bits. While the paper claims such occurrences are rare, this assumption may not hold when messages are short or divided into small blocks (as in MajorMark+). For example, when r=2 and b = 8, approximately 23.4% of the code space becomes infeasible, which is non-negligible.
- The system’s ability to distinguish between watermarked and non-watermarked text—crucial for LLM watermarking—is insufficiently evaluated. Although Appendix A.12 briefly discusses false positives, it omits key experimental details such as the threshold selection procedure and evaluation setup. Detection accuracy also seems to depend on the specific embedded message. Integrating and analyzing the detection method as part of the main algorithm would strengthen the work. In addition, this problem is a critical problem actively studied by the literature.
- The claim that a larger green-list ratio preserves text quality is overstated. While a larger ratio may help preserve fluency, it does not inherently ensure quality. The evaluation primarily relies on perplexity (PPL), which is limited in capturing semantic or stylistic fidelity. Additional text-quality metrics and qualitative comparisons would provide a more comprehensive assessment. Moreover, further justification is needed for using the top-5 hit rate as a key metric, as it could be trivially optimized by restricting watermarking to the top-5 tokens.
- Several experimental details are insufficiently described. For instance, when evaluating robustness against copy-paste and paraphrasing attacks, the ratio or extent of modified text should be explicitly reported.

**Questions:**

+ How is the embedded information selected during evaluation?

---

> ### Author Response · Authors · 2025-11-22
> **Responses to Reviewer tnCJ (1/5)**
>
> Dear reviewer tnCJ,
>
> **We sincerely appreciate your time in carefully reading our work and providing comments and suggestions to improve the quality of the manuscript. Below, we address your questions and concerns.**
>
> ---
>
> > **[W1] The paper claims that previous methods suffer from high computational complexity and that the proposed approaches are computationally efficient. However, the work lacks a detailed comparison or empirical analysis to substantiate this claim. In practice, the proposed methods appear to introduce additional computational costs due to multiple rounds of hashing, clustering-based decoding, and trial-and-error decoding processes.**
>
> **Response.** We thank the reviewer for raising this practical question regarding the computational complexity of our methods. We measured the average encoding and decoding time for generating/processing 500 tokens with message length $b=32$ and $\delta=2$ and provide a detailed analysis as follows.
>
> Table R1: Computational time comparison.
> | Method       | Encoding Time (s) | Decoding Time (s) | BA % |
> |-|-|-|-|
> | DepthW       | 37.87 | $\approx8.89\times 10^8$               | - |
> | MPAC         | 36.13 | 0.13                      | 89.22 |
> | RSBH         | 36.06 | 25.56                     | 89.38 |
> | MajorMark    | 36.58 | 2.18                      | 96.74 |
> | MajorMark+ (r=2)   | 36.85 | 12.70                     | 97.81 |
>
>
> 1. Encoding: Our encoding process introduces negligible overhead compared to standard generation. The only additional step is partitioning the permuted vocabulary, which is an $O(|\mathcal{V}|)$ operation.
>
> 2. Decoding: MajorMark performs two passes over the suspect text and applies a clustering procedure, resulting in a slightly increased overhead relative to MPAC (2.18 s vs. 0.13 s). MajorMark+, which requires $(b - r)$ decoding passes, achieves a total decoding time of 12.70 s, several orders of magnitude faster than exponential-time methods such as DepthW ($\approx 8.89 \times 10^8$ s), and also faster than RSBH (25.56 s). Although it is slower than the single-pass MPAC (0.13 s), the overall runtime remains well within a practical range.
>
> Importantly, as shown in the table, our methods deliver **significantly higher decoding accuracy** than existing approaches. This demonstrates a favorable property of our framework: it introduces only a modest computational cost while providing substantially improved robustness and reliability, making MajorMark and MajorMark+ far more effective for real-world watermark verification.
>
> We thank the reviewer for raising this practical question, which has helped us improve the completeness of our manuscript. We have incorporated the above analysis into Section 4.2 of the revised manuscript.
>
> ---

---

> ### Author Response · Authors · 2025-11-22
> **Responses to Reviewer tnCJ (2/5)**
>
> > **[W2] The decoding accuracy heavily depends on the specific embedded information. When the numbers of majority and minority bits are similar, decoding accuracy tends to be higher. In contrast, when the embedded bits are highly imbalanced (e.g., significantly more 1s than 0s), the variance in token distribution across shards diminishes, making decoding less effective. For instance, embedding 4 bits into 16 tokens with the message 1100 may yield shard distributions of {8, 8, 0, 0}, while 1110 could result in {5, 5, 6, 0}, illustrating that the decoding feature, variance in shard token counts, depends on the specific bit pattern.**
>
>
> **Response.** We thank the reviewer for this careful and insightful observation.
>
> 1. [**Imbalance leads to dilution.**] Mathematically, when the message is highly imbalanced (i.e., $h_\lambda \to b$), the green list ratio $\gamma$ approaches $1.0$. This means the "probability boost" provided by the watermark bias $\delta$ must be distributed across a much larger number of green shards. Consequently, the per-shard probability gain diminishes, reducing the mean gap between green and red shards.
>
> 2. [**Counter-mechanism: safety in high bias.**] However, this scenario creates a unique advantage that neutralizes the limitation. Because the green list covers the vast majority of the vocabulary (e.g., $\gamma > 80\%$) in these extreme cases, the "red list" effectively becomes a small set of tokens. This allows us to safely increase the watermark bias $\delta$ to much higher levels without degrading text quality.
>
> 3. [**Empirical verification.**] To validate this, we conducted experiments on MajorMark+ using "extreme" messages where the majority bit always occupies $>80\%$ of the message ($b=32$).
>
>     - By simply increasing $\delta$ to 4, MajorMark+ achieves a Bit Accuracy of 97.19% and a PPL of 4.36. This is fully comparable to the performance on standard balanced messages with $\delta=2$ (BA: 97.81%, PPL: 4.49).
>
> This demonstrates that robustness is not fundamentally compromised by bit patterns. The system effectively compensates for bit imbalance by leveraging the higher $\gamma$ to accommodate a stronger $\delta$, maintaining both high accuracy and text quality.
>
> We thank the reviewer for raising this important point, which helps improve the quality of our work. We have added this analysis to Appendix A.20 in the revised paper.
>
> ---
>
> > **[W3] The proposed system cannot embed messages consisting entirely of 0s or 1s, or containing long substrings of identical bits. While the paper claims such occurrences are rare, this assumption may not hold when messages are short or divided into small blocks (as in MajorMark+). For example, when r=2 and b = 8, approximately 23.4\% of the code space becomes infeasible, which is non-negligible.**
>
> **Response.** We thank the reviewer for this thoughtful observation. We acknowledge that both MajorMark and MajorMark+ exclude a small portion of the code space, as extreme messages consisting entirely of 0s or 1s lead to $\gamma=1.0$ and thus carry no watermark signal. However, this does not affect practical usability.
>
> 1. In our system design, watermark embedding is performed by the service provider, who can simply disable these extreme codes during encoding.
> 2. For MajorMark, only two codes (all-0 and all-1) are excluded—an entirely negligible fraction of the space.
> 3. As stated in our paper (Lines 322-327), the purpose of MajorMark+ is to mitigate the convergence of $\gamma$ toward 0.5 as $b$ grows, making it more suitable for long-bit messages (i.e., large $b$).
> 4. Moreover, when $b$ is small (e.g., $b = 8$), the total code space contains only 256 messages, which is far below what is required for a practical system requiring watermarking. For meaningful watermark messages (e.g., b $\geq$ 16), the excluded codes represent an insignificant portion of the overall code space for both MajorMark and MajorMark+.
>
> Therefore, the impact of the infeasible codes in both methods is negligible for all practical deployments.
>
> ---

---

> ### Author Response · Authors · 2025-11-22
> **Responses to Reviewer tnCJ (3/5)**
>
> > **[W4] The system’s ability to distinguish between watermarked and non-watermarked text—crucial for LLM watermarking—is insufficiently evaluated. Although Appendix A.12 briefly discusses false positives, it omits key experimental details such as the threshold selection procedure and evaluation setup. Detection accuracy also seems to depend on the specific embedded message. Integrating and analyzing the detection method as part of the main algorithm would strengthen the work. In addition, this problem is a critical problem actively studied by the literature.**
>
> **Response.**  We thank the reviewer for this important question regarding the decision boundary for false positive detection.
>
> 1. Empirical selection of $\tau = 2$. The threshold $\tau = 2$ was selected based on the empirical observation of the difference in standard deviations between the two majority bit hypotheses.
>
>     - For watermarked text: The shard counts for the correct majority bit $\lambda$ exhibit a high standard deviation (due to the boosted green shards), while the incorrect hypothesis yields a low standard deviation (uniform-like). This creates a large gap.
>     - For unwatermarked text: Both hypotheses yield random, uniform-like distributions with low standard deviations, resulting in a near-zero gap. We chose $\tau=2$ as a conservative margin to separate these two regimes under our default setting ($b=32$).
>
> 2. Sensitivity to message length ($b$). We evaluated the sensitivity of $\tau$ by varying $b$. In the table below, we report the standard deviations under different settings.
>
>     -Definitions: $\sigma_{\lambda}$ and $\sigma_{1-\lambda}$ denote the standard deviations of shard counts calculated under the hypotheses of the correct majority bit and the minority bit, respectively. For unwatermarked text, $\sigma_A$ and $\sigma_B$ simply represent the standard deviations corresponding to the arbitrary $\lambda=0$ and $\lambda=1$ hypotheses.
>
>     As shown in Table R1, as $b$ decreases (from 32 to 8), the number of shards decreases. This causes the expected number of tokens per shard ($T/b$) to increase significantly. The standard deviations naturally become larger. Consequently, a fixed $\tau=2$ becomes too strict, leading to higher FPR.
>
>     - Adaptation: To counteract this, we simply calibrate $\tau$ for lower $b$ (e.g., $\tau=4$ for $b=16$, $\tau=10$ for $b=8$). As shown in the table, this adjustment restores the FPR to 0% while maintaining 100% TPR, demonstrating that the metric remains discriminative even if the threshold value shifts.
>
>
>     Table R2: Sensitivity to message length $b$ ($T=500$)
>     | Message Length $b$       | Watermarked ($\sigma_{\lambda}$, $\sigma_{1-\lambda}$)| Unwatermarked  ($\sigma_{A}$, $\sigma_{B}$)| TPR / FPR ($\tau = 2$) | Calibrated TPR / FPR |
>     |-|-|-|-|-|
>     | 32 | (8.68, 4.34)  | (4.41, 4.57) | 100 / 5 |  - |
>     | 24 | (10.55, 4.86) | (5.20, 5.43) | 95 / 0 | - |
>     | 16 | (16.22, 6.08) | (6.08, 6.34) | 100 / 20 | 100 / 0 ($\tau = 4$)|
>     | 8  | (28.26, 7.81) | (8.22, 8.60) | 100 / 55 | 100 / 0 ($\tau = 10$)|
>
> 3. Sensitivity to text length ($T$).
> We further analyzed the impact of available tokens $T$ (with fixed $b=32, \tau=2$), as shown in Table R2.
>
>     - Observation: Reducing $T$ makes it harder to reconstruct the full shard distribution, slightly lowering the TPR (from 100% to 80%).
>     - Robustness: Crucially, the FPR remains at 0% even for short texts. This indicates that our method is "safe"—it may miss a watermark in very short texts (false negative), but it rarely falsely accuses unwatermarked text (false positive), which is the preferred behavior for user safety.
>
>     Table R3: Sensitivity to text length $T$ ($b=32, \tau=2$)
>     | Text Length $T$      | TPR | FPR |
>     |-|-|-|
>     | 500 | 100 | 5 |
>     | 450 | 100 | 0 |
>     | 400 | 90  | 0 |
>     | 350 | 80  | 0 |
>
> 4. Integration into the main algorithm. We agree that detection is an integral part of the system. We have formally incorporated this false positive detection step in the decoding pipeline of the revised manuscript. The decoder now checks the standard deviation gap against $\tau$ before attempting to extract the message bits. We leave the false positive detection module design for MajorMark+ as important future work.
>
> 5. We acknowledge that detection is actively studied in zero-bit watermarking. However, we respectfully note that this aspect is frequently overlooked in *multi-bit watermarking* literature. Major baselines like CycleShift, DepthW, and RSBH do not include mechanisms or even a discussion for false positive rejection. Only the very recent StealthInk explicitly addresses this.
>
> We sincerely thank the reviewer for your question again, which helps us improve the completeness of our manuscript. We have added these analyses to Appendix A.11 and refined the algorithm of MajorMark in the revised manuscript.
>
> ---

---

> ### Author Response · Authors · 2025-11-22
> **Responses to Reviewer tnCJ (4/5)**
>
> > **[W5] The claim that a larger green list ratio preserves text quality is overstated. While a larger ratio may help preserve fluency, it does not inherently ensure quality. The evaluation primarily relies on perplexity (PPL), which is limited in capturing semantic or stylistic fidelity. Additional text-quality metrics and qualitative comparisons would provide a more comprehensive assessment. Moreover, further justification is needed for using the top-5 hit rate as a key metric, as it could be trivially optimized by restricting watermarking to the top-5 tokens.**
>
> **Response.** We thank the reviewer for raising this important point regarding the evaluation metrics used in our work.
>
> 1. [**Clarification on "text quality".**] We respectfully clarify that in the context of our paper, "preserving text quality" refers to distributional fidelity—i.e., minimizing the divergence between the watermarked output and the original, unwatermarked distribution of the LLM. Our goal is to ensure the model's sampling behavior remains as close as possible to its natural state.
>
> 2. [**Additional semantic and lexical metrics.**]  We agree that PPL alone does not capture semantic preservation. To address this, we evaluated BERTScore (semantic similarity) and ROUGE-1 / ROUGE-Lsum (lexical/structural overlap) between watermarked and unwatermarked texts generated from the same prompts.
>
>     Table R4: Semantic and lexical fidelity (BERTScore / ROUGE-1 / ROUGE-Lsum) comparison across varying bias $\delta$ with $b=32$. Higher is better.
>     | Method | $\delta=2$ | $\delta=4$ | $\delta=6$ |
>     |-|-|-|-|
>     | MPAC | 0.8376 / 0.33 / 0.29 | 0.8210 / 0.28 / 0.25 | 0.8105 / 0.26 / 0.22 |
>     | RSBH | 0.8384 / 0.35 / 0.31 | 0.8316 / 0.33 / 0.29 | 0.8234 / 0.30 / 0.26 |
>     | MajorMark | 0.8415 / 0.37 / 0.33| 0.8305 / 0.33 / 0.28 | 0.8269 / 0.31 / 0.28 |
>     | MajorMark+ | 0.8449 / 0.36 / 0.32 | 0.8381 / 0.34 / 0.29 | 0.8311 / 0.33 / 0.29 |
>
>     - Semantic fidelity: MajorMark$^+$ consistently achieves the highest BERTScore across all bias settings, indicating that it preserves the original semantic meaning more effectively than MPAC or RSBH.
>
>     - Lexical fidelity: The higher ROUGE scores indicate that our method maintains substantial overlap in word choice and phrasing with the unwatermarked baseline. This confirms that the large green list allows the model to retain its original stylistic choices, rather than being forced to select synonyms or suboptimal tokens to satisfy the watermark constraint.
>
> 3. [**Justification for Top-5 hit rate.**] Regarding the Top-5 hit rate, we emphasize that we do not restrict sampling to the top-5 tokens during generation (we use standard nucleus sampling). Thus, this metric is not trivially optimized by hard constraints. Instead, we use it as a diagnostic metric for perturbation magnitude.
>
>     - A watermarking method "hits" the Top-5 if the originally high-probability tokens happen to be in the Green List (and get boosted) or remain high-probability despite not being boosted. Therefore, a high Top-5 hit rate therefore validates our theoretical claim: a larger green list statistically minimizes the disruption to the model's preferred token choices.
>
> We thank the reviewer for these important comments, which help improve the comprehensiveness of our work. We have added these additional results to Appendix A.19 in the revised manuscript.
>
> ---
>
> > **[W6] Several experimental details are insufficiently described. For instance, when evaluating robustness against copy-paste and paraphrasing attacks, the ratio or extent of modified text should be explicitly reported.**
>
> **Response.** We thank the reviewer for highlighting the need for precise experimental specifications.
>
> We have clarified the attack settings as follows:
>
> - Copy-Paste attack: We set the modification ratio to 10%. Specifically, we simulate a scenario where 10% of the tokens in the watermarked text are replaced with non-watermarked content (randomly interleaved) while maintaining the total length.
> - Paraphrase attack: We utilize the Dipper model (dipper-paraphraser-xxl). To ensure a rigorous evaluation against strong semantic attacks, we set both the lexical diversity and order diversity parameters to 20.
>
> We have moved these key experimental details from the Appendix to Section 4.2 of the main text, utilizing the additional page allowance provided during the rebuttal period.
>
> ---

---

> ### Author Response · Authors · 2025-11-22
> **Responses to Reviewer tnCJ (5/5)**
>
> > **[Q1] How is the embedded information selected during evaluation?**
>
> **Response.** We thank the reviewer for this practical question regarding our experimental setup. In our evaluation, the binary messages are randomly generated, assuming a uniform distribution. We explicitly exclude infeasible messages during generation. This selection process is fully implemented in our provided code (specifically the `load_or_generate_base_code()` function in `utils.py`) to ensure reproducibility.
>
> ---
>
> **We sincerely thank the reviewer again for your time and constructive comments. We hope these clarifications and additional results have fully resolved your concerns, and we remain open to any further discussion.**
>
> ---

---

### Official Review · Reviewer_FooY · 2025-10-31

**Soundness:** 3
**Presentation:** 3
**Contribution:** 2
**Rating:** 2
**Confidence:** 4

**Summary:**

This paper proposes an LLM watermarking method that leverages the majority bit and uses a clustering-based decoding strategy to improve the watermarking performance. The method recovers the embedded message by analyzing the occurrence of tokens across predefined vocabulary shards, enabling more accurate decoding.

**Strengths:**

1. The proposed method avoids the trade-offs as in prior works to improve the watermarking performance.

2. Experimental results show the performance is superior to some baselines.

**Weaknesses:**

1. The method is mainly compared to two prior works. Recent and stronger baselines are missing, such as UPV, SIR, SimMark, SemStamp, etc.

2. The method is largely a statistical refinement of existing token-level schemes, which limits its novelty.

3. Although the results show resistance against modification and paraphrasing attacks, it is unclear from a methodology perspective how the method can help improve the robustness. In addition, performance against stronger attackers also needs to be evaluated.

4. Complexity and overhead are only briefly discussed, which need more comprehensive evaluations.

**Questions:**

1. How do the methods compare to more recent works as listed above?

2. What is the computational complexity compared to prior works?

3. It also needs to be presented in greater detail how the majority bits are computed.

---

> ### Author Response · Authors · 2025-11-22
> **Responses to Reviewer FooY (1/4)**
>
> Dear Reviewer FooY,
>
> **We sincerely appreciate your time in carefully reading our work and providing comments and suggestions to improve the quality of the manuscript. Below, we address your questions and concerns.**
>
> ---
>
> > **[W1] The method is mainly compared to two prior works. Recent and stronger baselines are missing, such as UPV, SIR, SimMark, SemStamp, etc.**
>
> > **[Q1] How do the methods compare to more recent works as listed above?**
>
> **Response.** We thank the reviewer for this insightful suggestion and for bringing these important works to our attention. In our paper, we compare our methods against four state-of-the-art baselines (DepthW, CycleShift, MPAC, and RSBH), which represent the current leading approaches in **multi-bit watermarking**, the problem setting of interest in our work.
>
> Regarding the specific methods mentioned (UPV, SIR, SimMark, SemStamp), we carefully analyzed them and found that they primarily fall under the category of zero-bit watermarking (or detection-only watermarking):
>
> 1. UPV [1] and SIR [2] are robust token-level methods.
>
> 2. SimMark [3] and SemStamp [4] focus on sentence-level semantic watermarking.
>
> The fundamental goal of these methods is to output a binary decision (watermarked vs. non-watermarked), whereas our work targets multi-bit watermarking, which aims to embed and extract complex information (e.g., a 32-bit or 64-bit message). Consequently, these methods cannot be evaluated using bit accuracy, the primary metric for our task, making a direct quantitative comparison infeasible.
>
> We agree that discussing these works provides a more comprehensive view of the field. We have incorporated UPV, SIR, SimMark, and SemStamp into Section 2 to clarify the distinction between robust zero-bit detection and the multi-bit encoding problem we addressed.
>
> Reference:
>
> [1] An unforgeable publicly verifiable watermark for large language models, Liu et al., ICLR 2024.
>
> [2] A Semantic Invariant Robust Watermark for Large Language Models, Liu et al., ICLR 2024.
>
> [3] SimMark: A Robust Sentence-Level Similarity-Based Watermarking Algorithm for Large Language Models, Dabiriaghdam and Wang, Preprint 2025.
>
> [4] SemStamp: A Semantic Watermark with Paraphrastic Robustness for Text Generation, Hou et al., NAACL 2024.
>
> ---

---

> ### Author Response · Authors · 2025-11-22
> **Responses to Reviewer FooY (2/4)**
>
> > **[W2] The method is largely a statistical refinement of existing token-level schemes, which limits its novelty.**
>
> **Response.** We sincerely thank the reviewer for this feedback and would like to restate the novelty of our work more explicitly. We argue that MajorMark and MajorMark+ introduce a **fundamental paradigm shift** in how a multi-bit message is encoded and decoded, specifically designed to break the theoretical limits of prior works.
>
> 1. [**Existing green list hit-counting methods.**] Existing multi-bit watermarking schemes (e.g., MPAC, RSBH, CycleShift, DepthW) encode candidate binary messages into *green list hitting frequencies*, a design we refer to as the *green list hit-counting* paradigm. Specifically, after permuting the vocabulary, these methods randomly select a fixed $\gamma$ fraction of tokens as the green list, and their decoders rely on *counting* how many generated tokens fall into the recovered green list. As discussed in Lines 165–196 in our manuscript, this paradigm imposes a fundamental trade-off: to ensure a statistically detectable signal, the green list ratio $\gamma$ must be kept **small** ($\leq 0.5$), but a small $\gamma$ inevitably harms text quality.
>
> 2. [**Our green shard identity recovering method.**] In contrast, our methods (MajorMark and MajorMark+) directly *encode the message into token shard identities* rather than green list hitting frequencies. Specifically, instead of choosing a fixed green list, our majority bit–aware encoder partitions the permuted vocabulary into $b$ shards, one per message bit. Only shards corresponding to the majority bit are treated as green shards, and boosted by the watermark bias $\delta$. As a result, the union of green shards constitutes the green list of our methods. Because the majority bit always appears at least $b/2$ times in the message, the green list produced by our methods always covers *at least half of the vocabulary*. This inherently large green list ratio allows our approach to better preserve text quality. Although green list hit-counting methods could try to enlarge their green lists by manually increasing $\gamma$, Figure 1 in our manuscript shows that doing so severely degrades their decoding accuracy.
>
> 3. [**Decoupling watermarking signal srength from green list size.**] The most critical benefit of our majority bit-aware encoding is that it effectively decouples the watermarking signal strength from the green list size. In our framework, the watermark signal is defined by the identity of the specific token shards rather than an aggregate count. Consequently, the decoding goal shifts to recovering which shards were green during encoding. This mechanism achieves high decoding accuracy even when a large green list is used, a scenario where traditional counting-based methods fail.
>
> This shift, from encoding candidate message into green list hitting frequencies to shard identity, is the core novelty of our work. It **breaks the fundamental trade-off** faced by prior methods. Because our watermarking signal no longer depends on maintaining a small $\gamma$, we are able to do what was not feasible under the previous paradigm: use a *large* green list ($\gamma \ge 0.5$) to preserve text quality while simultaneously maintaining a strong, statistically recoverable watermarking signal. Thus, MajorMark and MajorMark+ represent not just refinements but a new watermarking framework specifically designed to resolve this long-standing limitation.
>
> We thank the reviewer for highlighting this point, which has allowed us to clarify the core novelty of our method. We have revised the manuscript (Lines 197-202) to more explicitly state this fundamental difference.
>
> ---

---

> ### Author Response · Authors · 2025-11-22
> **Responses to Reviewer FooY (3/4)**
>
> > **[W3] Although the results show resistance against modification and paraphrasing attacks, it is unclear from a methodology perspective how the method can help improve the robustness. In addition, performance against stronger attackers also needs to be evaluated.**
>
> **Response.** We thank the reviewer for raising this question about how our methods remain robust under copy-paste and paraphrasing attacks. The robustness of MajorMark and MajorMark+ stems from the novel **majority bit–aware encoding scheme**, which naturally yields a large green list and therefore provides resilience against both copy–paste and paraphrasing attacks. Below, we explain in detail why this scheme maintains the decoding accuracy of our methods, using paraphrasing attacks as an example.
>
> 1. Survival tokens: Paraphrasing, even when strong, does not replace all watermarked tokens. Each token that remains continues to vote correctly for its shard, which preserves part of the watermarking signal.
>
> 2. Replaced tokens: Unlike prior methods with small $\gamma$, our methods guarantee $\gamma \ge 0.5$ (and typically higher, $0.5 + 1/\sqrt{2\pi b}$ and $0.5 + 1/\sqrt{2\pi (b/r)}$). This means any "randomly replaced" token has a $\ge 50\%$ probability of falling into a green shard such that replacements dilute the signal rather than erase it.
>
> 3. Net effect: Consequently, the noise introduced by attacks tends to be neutral (diluting the distribution rather than erasing it) or even slightly biased toward the watermark (in imbalanced cases where $h_\lambda > b/2$). Our decoding algorithms remain stable when operating on this noisy but structurally preserved distribution.
>
> Overall, the majority bit–aware design ensures that both surviving and replaced tokens continue to support a stable watermarking signal, which allows our decoders to recover the message even under strong paraphrasing. We thank the reviewer again for this helpful comment, and we have added this discussion to Appendix A.18 in the revised paper.
>
> Regarding the evaluation against stronger attackers, we respectfully highlight that our experiments currently evaluate strong Copy-Paste and Paraphrasing attacks, which are commonly considered in previous literature (e.g., MPAC and RSBH). That said, we fully agree that the landscape of attacks is evolving. If the reviewer has specific other attack models or settings in mind, we would be grateful for the references and would be happy to discuss their potential impact on our method.
>
> ---
>
> > **[W4] Complexity and overhead are only briefly discussed, which need more comprehensive evaluations.**
>
> > **[Q2] What is the computational complexity compared to prior works?**
>
> **Response.** We thank the reviewer for raising this practical question regarding the computational complexity of our methods. We measured the average encoding and decoding time for generating/processing 500 tokens with message length $b=32$ and $\delta=2$ and provide a detailed analysis as follows.
>
> Table R1: Computational time comparison.
> | Method       | Encoding Time (s) | Decoding Time (s) | BA % |
> |-|-|-|-|
> | DepthW       | 37.87 | $\approx8.89\times 10^8$               | - |
> | MPAC         | 36.13 | 0.13                      | 89.22 |
> | RSBH         | 36.06 | 25.56                     | 89.38 |
> | MajorMark    | 36.58 | 2.18                      | 96.74 |
> | MajorMark+   | 36.85 | 12.70                     | 97.81 |
>
>
> 1. Encoding: Our encoding process introduces negligible overhead compared to standard generation. The only additional step is partitioning the permuted vocabulary, which is an $O(|\mathcal{V}|)$ operation.
>
> 2. Decoding: MajorMark performs two passes over the suspect text and applies a clustering procedure, resulting in a slightly increased overhead relative to MPAC (2.18 s vs. 0.13 s). MajorMark+, which requires $(b - r)$ decoding passes, achieves a total decoding time of 12.70 s, several orders of magnitude faster than exponential-time methods such as DepthW ($\approx 8.89 \times 10^8$ s), and also faster than RSBH (25.56 s). Although it is slower than the single-pass MPAC (0.13 s), the overall runtime remains well within a practical range.
>
> Importantly, as shown in the table, our methods deliver **significantly higher decoding accuracy** than existing approaches. This demonstrates a favorable property of our framework: it introduces only a modest computational cost while providing substantially improved robustness and reliability, making MajorMark and MajorMark+ far more effective for real-world watermark verification.
>
> We thank the reviewer for raising this practical question, which has helped us improve the completeness of our manuscript. We have incorporated the above analysis into Section 4.2 of the revised manuscript.
>
> ---

---

> ### Author Response · Authors · 2025-11-22
> **Responses to Reviewer FooY (4/4)**
>
> > **[Q3] It also needs to be presented in greater detail how the majority bits are computed.**
>
> **Response.** We thank the reviewer for this helpful comment. We clarify the computation of the majority bit below.
>
> Let $h_0$ and $h_1$ denote the number of occurrences of bits 0 and 1 in the message **m**, respectively. The **majority bit** in our method is determined by comparing $h_0$ and $h_1$: the bit with the larger count is designated as the majority bit. In the special case where $h_0 = h_1$, we set the majority bit $\lambda = 1$ by default.
>
> For example:
>
> - For **m = 10111011**, bit 1 appears more frequently than bit 0 ($h_1=6$ vs. $h_0=2$), so the majority bit is **1**.
>
> - For **m = 00110100**, bit 0 dominates ($h_0=5$ vs. $h_1=3$), so the majority bit is **0**.
>
> - For **m = 00001111**, where $h_0 = h_1 = 4$, we set $\lambda = 1$ by default.
>
> This simple rule illustrates how the majority bit–aware design captures the dominant statistical structure of the message, enabling robust decoding.
>
> ---
>
> **We sincerely thank the reviewer again for your time and constructive comments. We hope these clarifications and additional results have fully resolved your concerns, and we remain open to any further discussion.**
>
> ---

---

### Official Review · Reviewer_9MoG · 2025-11-01

**Soundness:** 3
**Presentation:** 3
**Contribution:** 3
**Rating:** 6
**Confidence:** 3

**Summary:**

This paper addresses the fundamental trade-off between text quality and decoding accuracy in multi-bit watermarking for Large Language Models (LLMs). The authors observe that existing methods suffer from this trade-off, which is governed by the size of the "green list" of preferred tokens. To overcome this, they propose MajorMark, a novel watermarking paradigm. The core idea is to construct the green list based on the majority bit λ of the message m, which theoretically guarantees a large green list ratio (γ ≥ 0.5) and thus preserves text quality. Crucially, MajorMark abandons traditional frequency-based decoding and instead employs a clustering-based strategy to recover the message by analyzing the distribution of token occurrences across vocabulary shards.
The paper further introduces MajorMark+, an enhanced version that partitions the message into blocks for encoding, leading to even better text quality. For decoding, MajorMark+ replaces clustering with a more robust deterministic decoding mechanism that enumerates possible majority bits and their counts, significantly improving decoding accuracy.

**Strengths:**

1.This paper reframes the multi-bit watermarking problem by decoupling the decoding process from the green list size. The "majority bit-aware" encoding and the subsequent distribution-based decoding (clustering/deterministic) represent a significant departure from prior art and a truly novel conceptual contribution.
2.This paper demonstrably achieves what it sets out to do: improve the trade-off between text quality and decoding accuracy. The experimental results across the board show that MajorMark+ in particular sets a new state-of-the-art, achieving lower perplexity (better quality) and higher bit accuracy simultaneously. This is a significant practical achievement.
3.This paper provides solid theoretical backing for its design choices, with formal proofs for the guaranteed green list size (γ ≥ 0.5) and its expected value. This combination of theoretical insight and empirical validation is the hallmark of a high-quality research paper.

**Weaknesses:**

1.While the paper correctly points out that the decoding complexity of MajorMark+ (b-r passes) is far more efficient than exponential methods, it is still a notable overhead compared to single-pass methods like MPAC. For a very long message b and small r, this could become a practical concern.
2.As shown in Figure 4, performance under strong paraphrase attacks degrades for all methods, including the proposed ones. While MajorMark and MajorMark+ are competitive, does the block-wise structure of MajorMark+ make it more vulnerable if a paraphraser alters the specific tokens used to select the block index?
3.The paper notes that extreme messages (all 0s or all 1s) lead to γ=1.0, rendering the watermark ineffective, and suggests disallowing them. While the practical impact is negligible for large b, this is a slight limitation in the universality of the codespace. For MajorMark+, this applies at the block level, which is a clever mitigation but still a constraint.

**Questions:**

1.Could you provide some empirical numbers on the actual wall-clock time required for decoding?
2.In MajorMark+, what's the reason for the performance drop under paraphrase attacks? Have you considered alternative, more robust ways of assigning tokens to blocks?
3.You propose a threshold on the standard deviation difference to detect unwatermarked text. How was this threshold (value of 2) chosen? Is it sensitive to the message length b or the text length T?

---

> ### Author Response · Authors · 2025-11-22
> **Responses to Reviewer 9MoG (1/3)**
>
> Dear reviewer 9MoG,
>
> **We sincerely appreciate your time in carefully reading our work and providing comments and suggestions to improve the quality of the manuscript. Below, we address your questions and concerns.**
>
> ---
>
> > **[W1] While the paper correctly points out that the decoding complexity of MajorMark+ ($b-r$ passes) is far more efficient than exponential methods, it is still a notable overhead compared to single-pass methods like MPAC. For a very long message b and small r, this could become a practical concern.**
>
> > **[Q1] Could you provide some empirical numbers on the actual wall-clock time required for decoding?**
>
> **Response.** We thank the reviewer for raising this practical question regarding the computational complexity of our methods. We measured the average encoding and decoding time for generating/processing 500 tokens with message length $b=32$ and $\delta=2$ and provide a detailed analysis as follows.
>
> Table R1: Computational time comparison.
> | Method       | Encoding Time (s) | Decoding Time (s) | BA % |
> |-|-|-|-|
> | DepthW       | 37.87 | $\approx8.89\times 10^8$               | - |
> | MPAC         | 36.13 | 0.13                      | 89.22 |
> | RSBH         | 36.06 | 25.56                     | 89.38 |
> | MajorMark    | 36.58 | 2.18                      | 96.74 |
> | MajorMark+ (r=2)   | 36.85 | 12.70                     | 97.81 |
>
>
> 1. Encoding: Our encoding process introduces negligible overhead compared to standard generation. The only additional step is partitioning the permuted vocabulary, which is an $O(|\mathcal{V}|)$ operation.
>
> 2. Decoding: MajorMark performs two passes over the suspect text and applies a clustering procedure, resulting in a slightly increased overhead relative to MPAC (2.18 s vs. 0.13 s). MajorMark+, which requires $(b - r)$ decoding passes, achieves a total decoding time of 12.70 s, several orders of magnitude faster than exponential-time methods such as DepthW ($\approx 8.89 \times 10^8$ s), and also faster than RSBH (25.56 s). Although it is slower than the single-pass MPAC (0.13 s), the overall runtime remains well within a practical range.
>
> Importantly, as shown in the table, our methods deliver **significantly higher decoding accuracy** than existing approaches. This demonstrates a favorable property of our framework: it introduces only a modest computational cost while providing substantially improved robustness and reliability, making MajorMark and MajorMark+ far more effective for real-world watermark verification.
>
> We thank the reviewer for raising this practical question, which has helped us improve the completeness of our manuscript. We have incorporated the above analysis into Section 4.2 of the revised manuscript.
>
> ---

---

> ### Author Response · Authors · 2025-11-22
> **Responses to Reviewer 9MoG (2/3)**
>
> > **[W2] As shown in Figure 4, performance under strong paraphrase attacks degrades for all methods, including the proposed ones. While MajorMark and MajorMark+ are competitive, does the block-wise structure of MajorMark+ make it more vulnerable if a paraphraser alters the specific tokens used to select the block index?**
>
> > **[Q2] In MajorMark+, what's the reason for the performance drop under paraphrase attacks? Have you considered alternative, more robust ways of assigning tokens to blocks?**
>
> **Response.** We thank the reviewer for these valuable comments and questions regarding the robustness of our methods under strong paraphrasing attack.
>
> 1. [**Reason for performance drop (signal dilution).**] The performance degradation under Dipper attacks, observed across all methods (including MPAC and RSBH), stems from the replacement of watermarked tokens. When a paraphraser substitutes a token, the new token is selected based on semantic similarity. Consequently, from the decoder's perspective, these replaced tokens act as random noise.
>
> 2. [**Robustness of block-wise structure.**] Modifying context tokens alters the computed block index $i$, potentially assigning a token to the wrong message block during decoding. We clarify that this does not create a specific structural vulnerability for the following reasons:
>     - Uniform noise: When the context is altered, the resulting block index $i$ becomes effectively random. This means the token's contribution is distributed randomly across all $r$ blocks rather than targeting and destroying a specific block.
>     - Graceful degradation: In MajorMark+, the message is recovered by aggregating statistics over the entire text. A "wrongly assigned" token simply fails to vote for the correct block (becoming neutral noise) rather than actively voting against it. Due to our high green list ratio, the remaining correctly synchronized tokens usually provide a sufficient signal to recover the embedded message.
>     - Independence: Since blocks are decoded independently, localized noise in one part of the text does not propagate to invalidate the decoding of other blocks.
>
> 3. [**Alternative assignment strategies.**] We fully agree with the reviewer that exploring paraphrase-invariant block assignment (e.g., mapping blocks based on sentence semantics or invariant features rather than sliding window tokens) is a promising direction. While we adopted the deterministic token-based mapping to maintain high embedding capacity and synchronization efficiency (standard in MPAC/RSBH), integrating robust semantic mapping could further mitigate decoding errors.
>
> We sincerely appreciate the reviewer’s comment, which helps improve the completeness of our manuscript. We have added this theoretical justification to Appendix A.18 in the revised version.
>
> ---
>
> > **[W3] The paper notes that extreme messages (all 0s or all 1s) lead to $\gamma$=1.0, rendering the watermark ineffective, and suggests disallowing them. While the practical impact is negligible for large b, this is a slight limitation in the universality of the codespace. For MajorMark+, this applies at the block level, which is a clever mitigation but still a constraint.**
>
> **Response.** We thank the reviewer for this thoughtful observation. We acknowledge that both MajorMark and MajorMark+ exclude a small portion of the code space, as extreme messages consisting entirely of 0s or 1s lead to $\gamma=1.0$ and thus carry no watermark signal. However, this does not affect practical usability.
>
> 1. In our system design, watermark embedding is performed by the service provider, who can simply disable these extreme codes during encoding.
> 2. For MajorMark, only two codes (all-0 and all-1) are excluded—an entirely negligible fraction of the space.
> 3. As stated in our paper (Lines 322-327), the purpose of MajorMark+ is to mitigate the convergence of $\gamma$ toward 0.5 as $b$ grows, making it more suitable for long-bit messages (i.e., large $b$).
> 4. Moreover, when $b$ is small (e.g., $b = 8$), the total code space contains only 256 messages, which is far below what is required for a practical system requiring watermarking. For meaningful watermark messages (e.g., b $\geq$ 16), the excluded codes represent an insignificant portion of the overall code space for both MajorMark and MajorMark+.
>
> Therefore, the impact of the infeasible codes in both methods is negligible for all practical deployments.
>
>
> ---

---

> ### Author Response · Authors · 2025-11-22
> **Responses to Reviewer 9MoG (3/3)**
>
> > **[Q3] You propose a threshold on the standard deviation difference to detect unwatermarked text. How was this threshold (value of 2) chosen? Is it sensitive to the message length b or the text length T?**
>
> **Response.** We thank the reviewer for this important question regarding the decision boundary for false positive detection.
>
> 1. Empirical selection of $\tau = 2$. The threshold $\tau = 2$ was selected based on the empirical observation of the difference in standard deviations between the two majority bit hypotheses.
>
>     - For watermarked text: The shard counts for the correct majority bit $\lambda$ exhibit a high standard deviation (due to the boosted green shards), while the incorrect hypothesis yields a low standard deviation (uniform-like). This creates a large gap.
>     - For unwatermarked text: Both hypotheses yield random, uniform-like distributions with low standard deviations, resulting in a near-zero gap. We chose $\tau=2$ as a conservative margin to separate these two regimes under our default setting ($b=32$).
>
> 2. Sensitivity to message length ($b$). We evaluated the sensitivity of $\tau$ by varying $b$. In the table below, we report the standard deviations under different settings.
>
>     - Definitions: $\sigma_{\lambda}$ and $\sigma_{1-\lambda}$ denote the standard deviations of shard counts calculated under the hypotheses of the correct majority bit and the minority bit, respectively. For unwatermarked text, $\sigma_A$ and $\sigma_B$ simply represent the standard deviations corresponding to the arbitrary $\lambda=0$ and $\lambda=1$ hypotheses.
>
>     As shown in Table R2, as $b$ decreases (from 32 to 8), the number of shards decreases. This causes the expected number of tokens per shard ($T/b$) to increase significantly. The standard deviations naturally become larger. Consequently, a fixed $\tau=2$ becomes too strict, leading to higher FPR.
>
>     - Adaptation: To counteract this, we simply calibrate $\tau$ for lower $b$ (e.g., $\tau=4$ for $b=16$, $\tau=10$ for $b=8$). As shown in the table, this adjustment restores the FPR to 0% while maintaining 100% TPR, demonstrating that the metric remains discriminative even if the threshold value shifts.
>
>
>     Table R2: Sensitivity to message length $b$ ($T=500$)
>     | Message Length $b$       | Watermarked ($\sigma_{\lambda}$, $\sigma_{1-\lambda}$)| Unwatermarked  ($\sigma_{A}$, $\sigma_{B}$)| TPR / FPR ($\tau = 2$) | Calibrated TPR / FPR |
>     |-|-|-|-|-|
>     | 32 | (8.68, 4.34)  | (4.41, 4.57) | 100 / 5 |  - |
>     | 24 | (10.55, 4.86) | (5.20, 5.43) | 95 / 0 | - |
>     | 16 | (16.22, 6.08) | (6.08, 6.34) | 100 / 20 | 100 / 0 ($\tau = 4$)|
>     | 8  | (28.26, 7.81) | (8.22, 8.60) | 100 / 55 | 100 / 0 ($\tau = 10$)|
>
> 3. Sensitivity to text length ($T$).
> We further analyzed the impact of available tokens $T$ (with fixed $b=32, \tau=2$), as shown in Table R3.
>
>     - Observation: Reducing $T$ makes it harder to reconstruct the full shard distribution, lowering the TPR (from 100% to 80%).
>     - Robustness: Crucially, the FPR remains at 0% even for short texts. This indicates that our method is "safe"—it may miss a watermark in very short texts (false negative), but it rarely falsely accuses unwatermarked text (false positive), which is the preferred behavior for user safety.
>
>     Table R3: Sensitivity to text length $T$ ($b=32, \tau=2$)
>     | Text Length $T$      | TPR | FPR |
>     |-|-|-|
>     | 500 | 100 | 5 |
>     | 450 | 100 | 0 |
>     | 400 | 90  | 0 |
>     | 350 | 80  | 0 |
>
>
> We sincerely thank the reviewer for your question again, which helps us improve the completeness of our manuscript. We have added these analyses to Appendix A.11 of the revised manuscript.
>
> ---
>
> **We sincerely thank the reviewer again for your time and constructive comments. We hope these clarifications and additional results have fully resolved your concerns, and we remain open to any further discussion.**
>
> ---

---

### Official Review · Reviewer_S6tj · 2025-11-01

**Soundness:** 3
**Presentation:** 2
**Contribution:** 2
**Rating:** 4
**Confidence:** 4

**Summary:**

This paper proposes MajorMark and MajorMark+, majority-aware multi-bit watermarking methods for large language models that replace frequency-based decoding with clustering, achieving higher decoding accuracy and better text quality without tuning the green-list ratio γ.

**Strengths:**

1. It relies on green-list frequency decoding, which inherently struggles to balance text quality and decoding accuracy.
2. The adoption of multi-bit block encoding is appropriate but not conceptually novel.
3. The clustering-based decoding design is adaptive and somewhat innovative, yet its theoretical foundation remains limited.

**Weaknesses:**

1. The decoding stage relies heavily on unsupervised clustering (K-Means) over vocabulary shards. However, no justification is provided for its stability under diverse token distributions, nor for convergence guarantees or error bounds.
2. The claimed robustness of MajorMark under paraphrasing attacks is demonstrated only empirically, without a clear mechanism or theoretical explanation for why clustering-based decoding should retain signal stability after semantic rewrites.
3. Only claimed results of LLaMA2-7b, other experimented SOTA LLMs should be shown and claimed.

**Questions:**

1. Please show more accurate SOTA-Based LLM results
2. Could the authors formalize why cluster separability is expected in the latent frequency space, rather than merely observed empirically?

---

> ### Author Response · Authors · 2025-11-22
> **Responses to Reviewer S6tj (1/3)**
>
> Dear reviewer S6tj,
>
> **We sincerely appreciate your time in carefully reading our work and providing comments and suggestions to improve the quality of the manuscript. Below, we address your questions and concerns.**
>
> ---
>
> > **[S1] It relies on green list frequency decoding, which inherently struggles to balance text quality and decoding accuracy.**
>
> > **[S2] The adoption of multi-bit block encoding is appropriate but not conceptually novel.**
>
> **Response.** We sincerely thank the reviewer for this feedback and would like to restate the novelty of our work more explicitly. We argue that MajorMark and MajorMark+ introduce a **fundamental paradigm shift** in how a multi-bit message is encoded and decoded, specifically designed to break the theoretical limits of prior works.
>
> 1. [**Existing green list hit-counting methods.**] Existing multi-bit watermarking schemes (e.g., MPAC, RSBH, CycleShift, DepthW) encode candidate binary messages into *green list hitting frequencies*, a design we refer to as the *green list hit-counting* paradigm. Specifically, after permuting the vocabulary, these methods randomly select a fixed $\gamma$ fraction of tokens as the green list, and their decoders rely on *counting* how many generated tokens fall into the recovered green list. As discussed in Lines 165–196 in our manuscript, this paradigm imposes a fundamental trade-off: to ensure a statistically detectable signal, the green list ratio $\gamma$ must be kept **small** ($\leq 0.5$), but a small $\gamma$ inevitably harms text quality.
>
> 2. [**Our green shard identity recovering method.**] In contrast, our methods (MajorMark and MajorMark+) directly *encode the message into token shard identities* rather than green list hitting frequencies. Specifically, instead of choosing a fixed green list, our majority bit–aware encoder partitions the permuted vocabulary into $b$ shards, one per message bit. Only shards corresponding to the majority bit are treated as green shards, and boosted by the watermark bias $\delta$. As a result, the union of green shards constitutes the green list of our methods. Because the majority bit always appears at least $b/2$ times in the message, the green list produced by our methods always covers *at least half of the vocabulary*. This inherently large green list ratio allows our approach to better preserve text quality. Although green list hit-counting methods could try to enlarge their green lists by manually increasing $\gamma$, Figure 1 in our manuscript shows that doing so severely degrades their decoding accuracy.
>
> 3. [**Decoupling watermarking signal strength from green list size.**] The most critical benefit of our majority bit-aware encoding is that it effectively decouples the watermarking signal strength from the green list size. In our framework, the watermark signal is defined by the identity of the specific token shards rather than an aggregate count. Consequently, the decoding goal shifts to recovering which shards were green during encoding. This mechanism achieves high decoding accuracy even when a large green list is used, a scenario where traditional counting-based methods fail.
>
> This shift, from encoding candidate message into green list hitting frequencies to shard identity, is the core novelty of our work. It **breaks the fundamental trade-off** faced by prior methods. Because our watermarking signal no longer depends on maintaining a small $\gamma$, we are able to do what was not feasible under the previous paradigm: use a *large* green list ($\gamma \ge 0.5$) to preserve text quality while simultaneously maintaining a strong, statistically recoverable watermarking signal. Thus, MajorMark and MajorMark+ represent not just refinements but a new watermarking framework specifically designed to resolve this long-standing limitation.
>
> We thank the reviewer for highlighting this point, which has allowed us to clarify the core novelty of our method. We have revised the manuscript (Lines 197-202) to more explicitly state this fundamental difference.
>
> As for the block-wise methods in MajorMark+, as we stated in Lines 322-327, the block encoding strategy is inspired by several previous works (e.g., MPAC, which is the first one using the block-wise encoding strategy), which is not a core contribution of our paper. This strategy helps MajorMark+ ensure a large green list when the message length $b$ is large.
>
> ---

---

> ### Author Response · Authors · 2025-11-22
> **Responses to Reviewer S6tj (2/3)**
>
> > **[S3] The clustering-based decoding design is adaptive and somewhat innovative, yet its theoretical foundation remains limited.**
>
> > **[W1] The decoding stage relies heavily on unsupervised clustering (KMeans) over vocabulary shards. However, no justification is provided for its stability under diverse token distributions, nor for convergence guarantees or error bounds.**
>
> > **[Q2] Could the authors formalize why cluster separability is expected in the latent frequency space, rather than merely observed empirically?**
>
> **Response.**  We thank the reviewer for highlighting the importance of providing a theoretical justification for the clustering-based decoding design in MajorMark. The separability of shard counts arises naturally from our encoding mechanism, which induces a bimodal Gaussian mixture distribution over the shard counts.
>
> 1. [**Theoretical justification.**] Under the random oracle assumption, the observed count $f_i$ of each shard $i$ can be well-approximated by a Gaussian distribution via the Central Limit Theorem. Specifically, the application of the watermark bias $\delta$ splits the shards into two groups with distinct expected sampling probabilities: $p_G$ for green shards (encoding the majority bit $\lambda$) and $p_R$ for red shards (encoding $1-\lambda$), where $p_G > p_R$. Consequently, the observed shard counts $f_i$ follow a mixture of two Gaussian distributions:$$f_i \sim
> \begin{cases}
> \mathcal{N}(\mu_G, \sigma_G^2), & \text{if shard } i \text{ encodes } \lambda \text{ (green)},\\\\
> \mathcal{N}(\mu_R, \sigma_R^2), & \text{if shard } i \text{ encodes } 1-\lambda \text{ (red)},
> \end{cases}$$where the expected counts are $\mu_G = T \cdot p_G$ and $\mu_R = T \cdot p_R$. Since KMeans is theoretically equivalent to a hard-assignment expectation-maximization algorithm for a Gaussian Mixture Model (GMM) with isotropic covariance, it is well-suited for this decoding task.
>
> 2. [**Empirical verification.**] This theoretical model is explicitly supported by our empirical results. As shown in our ablation study (Table 4 in the manuscript), we evaluated GMM as an alternative decoding algorithm. The results show that GMM achieves comparable BA to KMeans (e.g., 96.33% vs. 97.92% at $\delta=4$). This empirical success confirms that the underlying distribution of shard counts is indeed well-modeled as a mixture of Gaussians, validating both our theoretical analysis and the choice of clustering-based decoding.
>
> 3. [**Error analysis via distribution overlap.**] We further provide a theoretical error analysis to explain the boundary conditions of our decoding performance. Decoding errors occur when the clustering algorithm misclassifies a green shard (majority bit) as a red shard (or vice versa). Mathematically, this probability is governed by the overlap between the two Gaussian components $\mathcal{N}(\mu_G, \sigma_G^2)$ and $\mathcal{N}(\mu_R, \sigma_R^2)$.
>     - Error source 1: insufficient text length ($T$). When $T$ is small (e.g., extremely short texts), the variance is large relative to the mean gap. This results in significant distribution overlap, leading to a higher probability of clustering errors. This theoretical bound explains why BA drops in low-token regimes (as observed in Figure 6 of our paper).
>     - Error source 2: weak watermark bias ($\delta$). When $\delta$ is small (e.g., $\delta < 1.0$), the sampling probabilities $p_G$ and $p_R$ become nearly indistinguishable. Even with a large $T$, the means $\mu_G$ and $\mu_R$ remain too close, causing the two Gaussian modes to merge into a single unimodal distribution. In this regime, KMeans fails to find valid cluster centers, leading to random guessing. This aligns with our observation in Table 2, where a smaller $\delta$ yields lower accuracy than larger ones.
>
> We sincerely appreciate the reviewer’s comment, which helps improve the completeness of our manuscript. We have added this theoretical justification to Appendix A.17 in the revised version.
>
> ---

---

> ### Author Response · Authors · 2025-11-22
> **Responses to Reviewer S6tj (3/3)**
>
> > **[W2] The claimed robustness of MajorMark under paraphrasing attacks is demonstrated only empirically, without a clear mechanism or theoretical explanation for why clustering-based decoding should retain signal stability after semantic rewrites.**
>
> **Response.** We thank the reviewer for this insightful question. We agree that a theoretical mechanism is essential to explain the empirical robustness. The resilience of MajorMark stems from two coupled mechanisms: (1) the statistical nature of the noise introduced by paraphrasing, and (2) the stability of clustering-based decoding under such noise.
>
> 1. [**Nature of noise: watermarking signal dilution vs. destruction.**] Paraphrasing effectively replaces a subset of watermarked tokens with semantically similar ones.
>
>     - Survival tokens: Paraphrasing, even when strong, does not replace all watermarked tokens. Each token that remains continues to vote correctly for its shard, which preserves part of the watermarking signal.
>
>     - Replaced tokens: Unlike prior methods with small $\gamma$, our methods guarantee $\gamma \ge 0.5$ (and typically higher, $0.5 + 1/\sqrt{2\pi b}$). This means any "randomly replaced" token has a $\ge 50\%$ probability of falling into a green shard such that replacements dilute the signal rather than erase it.
>
>     - Net effect: Consequently, the noise introduced by attacks tends to be neutral (diluting the distribution rather than erasing it) or even slightly biased toward the watermark (in imbalanced cases where $h_\lambda > b/2$).
>
> 2. [**Clustering retains stability.**] Recall that our theoretical model where shard counts form a Bimodal Gaussian Mixture:$$f_i \sim \text{Mixture}(\mathcal{N}(\mu_G, \sigma_G^2), \mathcal{N}(\mu_R, \sigma_R^2)).$$
>
>     Paraphrasing reduces the effective watermarking signal, which shrinks the mean gap $\Delta\mu = \mu_G - \mu_R$. However, clustering-based decoding depends only on relative separability (bimodality). As long as the surviving signal maintains a non-zero gap $\Delta\mu$ sufficient to distinguish the two modes relative to their variance, KMeans naturally identifies the cluster centers—even when they have shifted significantly from their original positions. This adaptive property enables recovery.
>
>     Empirically, even under the strong paraphraser *Dipper* used in our evaluation, MajorMark consistently identifies the dominant shard signal within the perturbed token distribution, enabling robust decoding despite substantial noise.
>
> ---
>
> > **[W3] Only claimed results of LLaMA2-7b, other experimented SOTA LLMs should be shown and claimed.**
>
> > **[Q1] Please show more accurate SOTA-based LLM results.**
>
> **Response.** We thank the reviewer for this valuable suggestion on evaluating our methods on other LLMs. To demonstrate the generalizability of our approach, we have expanded our evaluation to include three additional SOTA open-source models: Qwen2.5-7B, Gemma-2B, and LLaMA-3.1-8B. We report the average BA (%) and PPL across watermarking biases $\delta \in \{2, 4\}$ for message lengths $b=32$ and $b=64$ in the tables below.
>
> Table R1: Average Performance (BA $\uparrow$ / PPL $\downarrow$) with $b=32$.
>
> | Model | MajorMark | MajorMark$^+$ |
> | :--- | :---: | :---: |
> | Qwen2.5-7B | 97.74 / 7.59 | 99.38 / 7.58 |
> | Gemma-2B | 97.74 / 8.41 | 99.69 / 8.32 |
> | LLaMA-3.1-8B | 97.27 / 6.82 | 98.91 / 6.68 |
>
> Table R2: Average Performance (BA $\uparrow$ / PPL $\downarrow$) with $b=64$.
>
> | Model | MajorMark | MajorMark$^+$ |
> | :--- | :---: | :---: |
> | Qwen2.5-7B | 92.78 / 7.81 | 95.63 / 7.66 |
> | Gemma-2B | 92.58 / 9.17 | 96.56 / 8.62 |
> | LLaMA-3.1-8B | 91.06 / 6.64 | 95.78 / 6.70 |
>
> These results confirm that both MajorMark and MajorMark$^+$ maintain high effectiveness and strong generalization capabilities across diverse model architectures. We have included these detailed results in Appendix A.6 of our manuscript.
>
> ---
>
> **We sincerely thank the reviewer again for your time and constructive comments. We hope these clarifications and additional results have fully resolved your concerns, and we remain open to any further discussion.**
>
> ---

---

### Author Response · Authors · 2025-11-22
**Official Comment by Authors**

Dear all reviewers,

We once again thank you for taking the time to read our paper carefully and for providing insightful comments and constructive suggestions that helped improve the quality of our work. We hope that our responses have addressed your concerns and questions. All revisions in the manuscript have been marked in **blue**.

*We are open to any further discussion.*

Regards,

Authors of submission 6174

---

### Author Response · Authors · 2025-12-03
**General Response to ACs and Senior ACs**

Dear Area Chairs and Senior Area Chairs,

We thank the Area Chairs for meta-reviewing our submission under the current special circumstances, and we sincerely appreciate the reviewers for their evaluation and constructive feedback.

---

Overall, the reviewers recognized both the methodological contribution and the empirical effectiveness of our approach, including its strong robustness and practical decoding performance, while raising questions mainly about robustness mechanisms, computational complexity, theoretical justification of clustering-based decoding, detection reliability, and evaluation completeness. We summarize how these concerns were addressed below.

---

Reviewer FooY questioned the missing comparison with recent watermarking baselines and the novelty of our method. We clarified that the suggested methods (UPV, SIR, SimMark, SemStamp) focus on zero-bit detection and are therefore not directly comparable to our multi-bit watermarking setting. We added these works to the Related Work for completeness. We also clarified that MajorMark introduces a new paradigm that encodes messages into shard identities rather than green-list hit counts, breaking the fundamental trade-off between text quality and decoding accuracy. In addition, we provided a detailed theoretical explanation and empirical justification for robustness under paraphrasing attacks, and added a full computational complexity analysis.

---

Reviewer S6tj raised concerns about the block encoding, the theoretical foundation and stability of clustering-based decoding, robustness under paraphrasing, and generalization to other LLMs. We clarified that block encoding is not claimed as our core contribution, and that our main novelty lies in the majority bit–aware shard identity framework. We provided a full theoretical justification showing that shard counts follow a bimodal Gaussian mixture, which explains the separability and stability of KMeans decoding. We further explained the robustness mechanism under paraphrasing via signal dilution rather than destruction. To demonstrate generalizability,  experiments on Qwen2.5-7B, Gemma-2B, and LLaMA-3.1-8B consistently show high BA and stable PPL.

---

Reviewer 9MoG focused on the decoding overhead of MajorMark+, robustness of block-wise decoding under paraphrase attacks, code-space limitations, and threshold-based detection. We provided concrete wall-clock decoding times and showed that MajorMark+ remains far more efficient than exponential baselines while achieving much higher decoding accuracy. We explained why block-wise decoding degrades gracefully under paraphrasing and does not introduce structural vulnerability. We clarified that the excluded extreme codes form a negligible portion of the practical code space. We also provided a detailed sensitivity analysis of the detection threshold with respect to message length and text length, and formally integrated the detection step into the decoding pipeline.

---

Reviewer tnCJ raised concerns about computational overhead, sensitivity to message imbalance, infeasible codes, detection reliability, and insufficient experimental details. We added a full runtime comparison against prior methods, showing that our decoding cost is practical relative to the achieved robustness gains. We theoretically analyzed why bit imbalance leads to signal dilution and showed that increasing the watermark bias compensates for this effect with negligible quality loss. We clarified the handling and practical impact of infeasible codes. We expanded the detection evaluation with threshold calibration, TPR/FPR analysis, and explicit integration into the algorithm. We also moved key attack settings and experimental details from the appendix into the main text for clarity.

---

Overall, we substantially strengthened the manuscript by adding new theoretical analyses, multiple new results, sensitivity studies, detailed runtime and detection analyses, and clearer experimental specifications. We believe that all major concerns raised by the reviewers have been fully addressed in the revised version, and that the current manuscript presents a more rigorous, complete, and well-justified multi-bit watermarking framework.

Best regards,

Authors of the Submission 6174

---

### Meta-Review · Area_Chair_MoWr · 2026-01-06

**Summary:**

The paper introduces MajorMark and MajorMark+, majority-bit–aware watermarking methods for LLMs aimed at improving the trade-off between text quality and decoding accuracy. While the idea is interesting and the topic is important, most reviewers raised concerns about limited novelty, reliance on clustering without strong theoretical guarantees, incomplete evaluation against recent baselines, and practical limitations such as computational overhead and robustness under paraphrasing. Only one reviewer leaned slightly positive, while others recommended rejection or were marginally negative.

**Reviewer Concerns:**

The rebuttal clarified some points, such as the distinction from zero-bit watermarking methods, theoretical justification for clustering, and added runtime comparisons. However, key concerns remain:

+ Novelty is still perceived as incremental by multiple reviewers.
+ Robustness and detection reliability lack comprehensive theoretical and empirical support.
+ Evaluation against stronger baselines and additional quality metrics is insufficient.
+ Practical limitations (e.g., decoding overhead, infeasible codes) persist.

Overall, the responses improved clarity but did not fully resolve the fundamental issues.

**Reviewer Scores:**

- Reviewer FooY (Reject): Likely unchanged; concerns about novelty and missing baselines remain.
- Reviewer tnCJ (Marginally below): May stay the same; rebuttal addressed details but not core limitations.
- Reviewer S6tj (Marginally below): Possibly unchanged; theoretical justification added but robustness concerns persist.
- Reviewer 9MoG (Marginally above): Might remain slightly positive; acknowledges improvements but still notes overhead and robustness issues.

---

### Decision · Program_Chairs · 2026-01-26

Reject